# Time-Varying Elasticity of Cyclically Adjusted Primary Balance and Effect of Fiscal Consolidation on Domestic Government Debt in South Africa

Eugene Msizi Buthelezi * and Phocenah Nyatanga

School of Accounting, Economics and Finance, University of KwaZulu-Natal, Pietermaritzburg 3219, KwaZulu-Natal, South Africa
* Correspondence: msizi1106@gmail.com

**Abstract:** This paper investigates the impact of the time-varying elasticity of the cyclically adjusted primary balance (CAPB) and fiscal consolidation on government debt. The time-varying parameter structural vector autoregression (TVP-VAR) model is used on a time series of data from 1979 to 2022. The contribution of this paper is on the understanding of the impact of fiscal consolidation on domestic government debt and the need to use time-varying elasticity when calculating the cyclical adjusted primary balance to provide a more accurate representation of discretionary actions taken by fiscal authorities. It is found that there is more variation in the CAPB with time-varying elasticity than with constant elasticity. Constant elasticity is not effective in capturing fiscal consolidation episodes, and time-varying elasticity is a better alternative. There is evidence that fiscal consolidation increases domestic government debt. The shocks of fiscal consolidation through government expenditure cuts reduce domestic government debt in the long run, while taxes increase domestic government debt. It is recommended that fiscal authorities use fiscal consolidation to reduce government expenditure that is related to inefficient expenditure. In the event of government expenditure, this expenditure needs to be in productive sectors of the economy that will bring about an increase in revenue rather than an increase in the tax rate. Given the result, a tax increase should be something that fiscal authorities are not using in the effort to stimulate economic growth or reduce domestic government debt.

**Keywords:** fiscal consolidation; domestic government debt; cyclically adjusted primary balance (CAPB); time-varying parameter structural vector autoregression (TVP-VAR)

**JEL Classification:** C82; E02; E60; E62

## 1. Introduction

The debate on fiscal consolidation and measures has been of interest to the Organisation for Economic Co-operation and Development (OECD), the Internal Monetary Fund (IMF), and other scholars (Alesina and Perotti 1995; Alesina and Perotti 1997; Alesina et al. 2015; Yang et al. 2015) among other. However, there is no agreement on the impact of fiscal consolidation on government debt. The thinking around fiscal consolidation is that government expenditure cuts and tax increases will result in a fall in government debt. This is because forward-looking economic agents will anticipate a reduction in tax and interest rates. This will increase permanent income, crowd-in investment, increase economic activities, and result in higher tax collection that can be used to reduce government debt (Alesina and Ardagna 2010; Mankiw 2019). One of the broad measures of discretionary government intervention to reduce the government debt that defines fiscal consolidation episodes is the cyclically adjusted primary balance (CAPB). This measure is concerned with the identification of discretionary fiscal policy changes in taxes and government expenditure by filtering out changes due to economic fluctuations in taxes as well as government expenditure (Alesina and Perotti 1995).

There are five main developments of the CAPB. The first is the primary approach, concerned with differences in government expenditure and taxes to find fiscal consolidation (Alesina and Perotti 1995). The second is the Blanchard approach, which considers cyclical components (government expenditure and tax) in the CAPB to find fiscal consolidation (Blanchard 1990). The third is the asset price approach, which filters out the asset price impact on government revenue and expenditure (Yang et al. 2015). Fourth and fifth are the OECD approach as well as the IMF approach, which uses deviation from the output gap from government expenditure and taxes to find fiscal consolidation. Both the OECD and IMF approaches use the constant elasticity of government expenditure and taxes to reflect the responsiveness required to find fiscal authorities' actions or the discretionary actions of fiscal authorities that can be attributed to fiscal consolidation espisodess (Zhang et al. 2022).

The OECD and IMF assume constant tax revenue and government expenditure elasticities throughout the analysis in the compositions of the CAPB (Mourre et al. 2013). These elasticities are used to identify discretionary actions by fiscal authorities that are to reduce domestic government debt through an increase in taxes and a cut in government expenditure. This paper identifies this as a problem because constant elasticity in the CAPB does not account for responsiveness to changes in these economic variables over time. The applicability of constant elasticity in government expenditure does not capture well government interventions over time or the fiscal framework that may be used to reduce domestic government debt. On the other hand, there is an economic dynamic in government revenue, and constant elasticity ignores every change in the tax system and taxes. In this regard, the assumption of constant elasticity in tax revenue and government expenditure categories by the OECD and IMF approaches can have significant biases. This leads to the incorrect composition of the CAPB as well as the identification of fiscal consolidation episodes.

In South Africa, domestic government debt was high at a rate of 71.72% in 2021 (SARB 2022). This rate of 71.72% is above the 60% threshold advocated by Southern African Development Community (SADC) countries, of which South Africa is one (Buthelezi and Nyatanga 2018). On the other hand, in 2014, the Financial and Fiscal Commission (FFC) recommended more fiscal consolidation stances to restore the fiscal position and reduce government debt (BR 2014). The FFC recommendation outlined that "Fiscal consolidation can no longer be postponed. Ensuring continued progress toward a better life obliges the government to safeguard public finances by acting within fiscal limits that can be sustained over the long term. To do otherwise would risk exposing the country to a debt trap, with damaging consequences for development for many years to come" (MTBPS 2014). The question this paper poses is as follows: what is the difference between the CAPB with time-varying elasticity and time-invariant elasticity as a proxy for fiscal consolidation? The second concern of this paper is the impact of fiscal consolidation on government debt in South Africa. The time-varying parameter structural vector autoregression (TVP-VAR) model following Nakajima (2011) was used on data from 1979 to 2022. The economic variables used were domestic government debt, money supply, total government revenue, debt service ratio, and fiscal consolidation, proxied by the CAPB, among others.

The significance of this paper lies in its contribution to the understanding of the impact of fiscal consolidation on domestic government debt and the need to use time-varying elasticity when calculating the cyclical adjusted primary balance (CAPB) to provide a more accurate representation of discretionary actions taken by fiscal authorities. The paper highlights the limitations of using constant elasticity when calculating the CAPB, which can result in an inaccurate representation of discretionary actions taken by fiscal authorities. The paper also demonstrates the significant impact of fiscal consolidation on government debt levels, particularly in the short term, and the need for careful planning and consideration of potential negative impacts on government debt levels. The paper provides important policy implications and recommendations for future research and policy decisions, including the need for the IMF and OECD to adopt time-varying elasticity when calculating the CAPB, the importance of examining the short-term benefits of fiscal consolidation, and the need to consider the impact of fiscal consolidation on other economic variables beyond government

debt levels. The paper contributes to the ongoing debate and discussion surrounding fiscal policy and its impact on government debt levels, providing important insights for policymakers and researchers alike.

It is found that the CAPB calculated with constant elasticity has a lower period compared with that calculated with time-varying elasticity. There is less variation in the CAPB with constant elasticity, while the CAPB with time-varying elasticity reflects more variation (more discretionary action) and covers 43 years. The IMF reflects that the CAPB has a non-upwards trend that reflects increasingly less discretionary action. However, when time-varying elasticity is accounted for, fiscal consolidation episodes are found. There is a 56.26% variation in the CAPB with time-varying elasticity, and there is a 2.36% variation in the CAPB of the IMF data. Fiscal consolidation government expenditure is found to increase domestic government debt, while tax-increase fiscal consolidation results in a slight fall in domestic government debt and quickly return to equilibrium. It is recommended that the IMF and OECD move away from constant elasticity and start to utilise time-varying elasticity to capture the fiscal consolidation variation over time. Fiscal consolidation has no positive impact on reducing domestic government debt. Fiscal authorities need to use government expenditure in productive sectors of the economy that will bring about an increase in revenue rather than an increase in the tax rate, as advocated in the fiscal consolidation policy. Moreover, fiscal authorities need to develop a tax system that generates optimal tax revenue with the adjustment of the tax rates.

The rest of this paper is outlined as follows. First, Section 2 outlines a review of fiscal consolidation measures. Second, Section 3 outlines an empirical review measure of fiscal consolidation. Third, Section 4 discusses this paper's methodology. Fourth, Section 5 discusses the empirical results. Finally, Section 6 outlines the conclusion of the paper.

## 2. Review of Fiscal Consolidation Measures

Scholars have been interested in how to measure fiscal consolidation, which reflects discretionary government intervention to reduce government debt (Alesina and Perotti 1995; Romer and Romer 2010; Leigh et al. 2011). The use of the CAPB has been proposed to identify discretionary fiscal policy changes in taxes and government expenditure by filtering out changes that are due to economic fluctuations (Alesina and Perotti 1995). There are five main developments of the CAPB, outlined below.

### 2.1. Primary Approach

The primary approach rationale is that the CAPB can be presented by the changes in the primary deficit as advocated by (Alesina and Perotti 1995). The primary approach is shown in Equation (1).

$$\Delta CAPB_t = (TGR_t - G_t) - (TGR_{t-1} - G_{t-1}) \tag{1}$$

where $TGR_t$ is total government revenue, $G_t$ is total government expenditure, and $t$ reflects the time. The thinking is that a positive difference between the current budget balance $(TGR_t - G_t)$ and the previous budget balance $(TGR_{t-1} - G_{t-1})$ reflects the discretionary actions of the government.

### 2.2. Blanchard Approach

Blanchard (1990) notes that economic variables that show changes or deviation from full employment are critical in the calculation of the CAPB. Blanchard (1990) points out that unemployment triggers cyclical movement in taxes and government expenditures. Unemployment changes were proposed to be filtered out in the CAPB calculation to find discretionary changes in fiscal policy that can be attributed to fiscal consolidation. The primary approach was developed into the Blanchard approach, as shown in Equation (2).

$$\Delta CAPB_t = (TGR_t - G_t(UNE_{t-1})) - (TGR_{t-1} - G_{t-1}) \tag{2}$$

where $UNE_{t-1}$ is unemployment for the prior year and other economic variables are as indicated above. The drawback of this approach is that unemployment in a country may be affected by external factors.

### 2.3. Asset Price Approach

The asset price approach affects the CAPB and needs to be filtered out in the CAPB. This is because a boom in the stock market improves the CAPB by increasing capital gains and cyclically adjusted tax revenues (Yang et al. 2015). The asset price approach advocates that asset prices need to be filtered out and fiscal authorities need to consider time trends as well as unemployment, as shown in Equation (3).

$$G_t = \alpha_0 + \alpha_1 Tremd + \alpha_2 UNE_t + e_t \tag{3}$$

where *Tremd* is the government expenditure and $e_t$ is the residuals. To find the primary adjusted government expenditure for business cycles and changes in unemployment, Equation (4) is followed.

$$G_t(UNE_{t-1}) = \alpha_0 + \alpha_1 Tremd + \alpha_2 UNE_t + e_t \tag{4}$$

where ($\alpha_0$, $\alpha_1$, $\alpha_2$) are estimated coefficients and the asset price index is added as in Equation (5).

$$TGR_t = \alpha_0 + \alpha_1 Tremd + \alpha_2 UNE_t + \alpha_3 Assetprice_t + e_t \tag{5}$$

where $Assetprice_t$ is the asset price index and the discretionary revenue changes as shown with the account of the previous year's asset price difference $(t-1)$, as shown in Equation (6).

$$TGR_t(UNE_{t-1}, Assetprice_{t-1}) = \alpha_0 + \alpha_1 Tremd + \alpha_2 UNE_t + \alpha_3 Assetprice_t + e_t \tag{6}$$

Finally, the changes in discretionary fiscal policy are obtained in Equation (7).

$$\Delta CAPB_t = (TGR_t(UNE_t, Assetprice_t) - G_t(UNE_t)) - (TGR_{t-1}(UNE_{t-1}, Assetprice_{t-1}) - G_{t-1}(UNE_{t-1})) \tag{7}$$

### 2.4. OECD Approach

The OECD approach focuses on the elasticity of government expenditures and taxes to find the discretionary action of fiscal consolidation. The OECD approach rationale is that discretionary changes are best presented when the present primary deficit would have prevailed if expenditure in the previous year had grown with potential GDP and revenues had grown with actual GDP (Mourre et al. 2013). The OECD approach is reflected in Equation (8).

$$\Delta CAPB_t = \frac{\left[ \left( \sum_{j=1}^{4} TGR_t^{\varepsilon_{tgr}} - G_t^{\varepsilon_g} \right) - \left( \sum_{j=1}^{4} TGR_{t-1}^{\varepsilon_{tgr}}(1+y_t) - G_{t-1}^{\varepsilon_g}(1+y_t) \right) \right]}{Y_{t-1}} \tag{8}$$

where $y$ is nominal GDP and $Y$ is the nominal GDP potential, which is estimated based on country-specific production functions. The OECD approach offers a much broader scope of the CAPB because it involves a disaggregated approach and the elasticity of $\varepsilon_{tgr}$ (tax revenue) and $\varepsilon_g$ (government expenditure) (Mourre et al. 2013). There are four tax revenue categories, which are shown in Equations (9)–(12).

$$\Delta CIT_t = \beta_0 + \beta_1 CIT_t \left( \frac{y_t}{Y_t} \right) + \mu_t \tag{9}$$

$$\Delta PIT_t = \beta_0 + \beta_1 PIT_t \left( \frac{y_t}{Y_t} \right) + \mu_t \tag{10}$$

$$\Delta SSC_t = \beta_0 + \beta_1 SSC_t \left( \frac{y_t}{Y_t} \right) + \mu_t \tag{11}$$

$$\Delta IT_t = \beta_0 + \beta_1 IT_t \left( \frac{y_t}{Y_t} \right) + \mu_t \tag{12}$$

where *CIT* is corporate income tax, *PIT* is personal income tax, *SSC* is social security contributions, and *IT* is indirect taxes. On the expenditure side, *umplb* is unemployment benefits, as shown in Equation (13).

$$\Delta \frac{UNE_t}{UNE_{t-1}} = \beta_0 + \beta_1 unmplb_t \left( \frac{y_t}{Y_t} \right) + \mu_t \tag{13}$$

where $UNE_t$ reflect the unemployment and $UNE_{t-1}$ is the unemployment in the last period while represents the $\Delta$ change. The CAPB reflects the cyclically adjusted tax revenue and cyclically adjusted government expenditure accounting for elasticity as well as the output gap, as shown in Equation (14).

$$\Delta CAPB_t = \sum_{j=1}^{4} TGR_t \left( \frac{y_t}{Y_t} \right)^{\varepsilon_r} - G_t \left( \frac{y_t}{Y_t} \right)^{\varepsilon_g} \tag{14}$$

The CAPB is derived from constant cyclically adjusted tax revenue $\varepsilon_{tgr}$ and government expenditure $\varepsilon_g$ accounting for elasticity as well as the output gap (Alesina and Perotti 1997; Alesina and Ardagna 2013; Alesina et al. 2015; Alesina et al. 2019). The OECD approach uses generalized least squares (GLS) to estimate the elasticity for each country and the seemingly unrelated regression procedure (SURE). This estimation is reflected in Equations (15)–(18).

$$\varepsilon_{tgr} \mid = \mid TGR_t = \beta_0 + \beta_1 \sum_{j=1}^{4} TGR_t \left( \frac{y_t}{Y_t} \right) + \mu_t \tag{15}$$

$$\varepsilon_{tgr} = \beta_1 = cnstnt\_elstcy\_tgr \tag{16}$$

$$\varepsilon_g \mid = \mid G_t = \beta_0 + \beta_1 G_t \left( \frac{y_t}{Y_t} \right) + \mu_t \tag{17}$$

$$\varepsilon_{tgr} = \beta_1 = cnstnt\_elstcy\_tgr \tag{18}$$

where *cnstnt_elstcy_tgr* and *cnstnt_elstcy_tgr* reflect the constant elasticity.

*2.5. International Monetary Fund Approach*

Similar to the OECD, the IMF follows a similar approach to finding the CAPB, as shown in Equation (19).

$$\Delta CAPB_t = \frac{\left[ \left( \sum_{j=1}^{4} TGR_t^{\varepsilon_{tgr}} - G_t^{\varepsilon_g} \right) - \left( \sum_{j=1}^{4} TGR_{t-1}^{\varepsilon_{tgr}} (1 + y_t) - G_{t-1}^{\varepsilon_g} (1 + y_t) \right) \right]}{Y_{t-1}} \tag{19}$$

The only difference is that the OECD uses GMM to find elasticity to obtain the potential output, while the IMF utilizes the Hodrick–Prescott (HP) filter, which is a data-smoothing technique, over all the data points (Mourre et al. 2013).

*2.6. Narrative Approach*

The narrative approach rationale is that historical documents that outline the intentions of fiscal authorities to increase taxes and reduce government expenditures are those that fully reflect discretionary changes by fiscal authorities that can be attributed to fiscal

consolidation (Romer and Romer 2010; Leigh et al. 2011). Romer and Romer (2010) and Leigh et al. (2011) build fiscal consolidation episodes, which are shown in Equation (20):

$$FC_t = FC_t^G + FC_t^T + \epsilon_t \tag{20}$$

where $FC_t$ is narrative fiscal consolidation episodes $FC_t^G$ is a government expenditure cut and $FC_t^T$ is a tax increase. The fiscal consolidation episode follows policy documents advocate that outline a tax increase and a cut in government expenditure.

### 2.7. Definition Approach

The definition approach is based on thresholds or specific changes in fiscal variables such as government debt, CAPB, and deficit (Bergman and Hutchison 2010). The intuition is that a fall in government debt, which is the ultimate objective of fiscal consolidation, best presents a discretionary action. The definition approach to the threshold is shown in Table 1.

**Table 1.** Definition approach to the threshold.

| Economic Variables | Fiscal Consolidation Definition |
| --- | --- |
| Government debts share to gross domestic product | A 4.5% decrease in government debt share to gross domestic product (GDP) in $(t+1)$, $(t+2)$, and $(t+3)$ (Alesina and Ardagna 2010). The Mean is less than 5% of the initial government debt share to GDP for 3 successive years (Alesina and Perotti 1995; Alesina and Ardagna 2010). |
| Government deficit | A fall of 2% below the initial rate for government deficit in $(t+1)$, $(t+2)$, and $(t+3)$ (Alesina and Perotti 1995; Alesina and Ardagna 2010). |
| Economic growth | Economic growth is higher for 2 consecutive years for the growth rate means of cases where there was fiscal consolidation (Alesina et al. 1998). The average economic growth rate, at $(t)$, is higher than $(t-1)$ and $(t-2)$ (Giudice and Turrini 2007). |
| The cyclically adjusted primary balance | If there is a 1% change in the cyclically adjusted primary balance over 3 years (Tavares 2004). The cyclically adjusted primary balance improves by 1.5% in $(t)$ (Alesina and Perotti 1997; Alesina et al. 1998; Gupta et al. 2005; Alesina and Ardagna 2010; Hernández De Cos and Moral-Benito 2013; Schaltegger and Weder 2014). The cyclically adjusted primary balance improves by 1.5% in $(t+1)$ and $(t+2)$ (Alesina et al. 1998). The cyclically adjusted primary balance increases by 2% in $(t+1)$ (Alesina et al. 1998). The cyclically adjusted primary balance improves by mean $(\mu)$ plus standard deviation $(\sigma)$ in $(t)$ (Yang et al. 2015). |

Composed by the authors.

## 3. Literature Review of Fiscal Consolidation Measures

Giorno et al. (1995) used the OECD methodology of CAPB elasticity from 1978 to 1992. The tax revenue elasticity function is $\varepsilon_{tgr} \mid = \mid TGR_t = \beta_0 + \beta_1 \sum_{j=1}^{4} TGR_t \left(\frac{y_t}{Y_t}\right) + \mu_t$ and the government expenditures categories are represented by $\varepsilon_g \mid = \mid G_t = \beta_0 + \beta_1 G_t \left(\frac{y_t}{Y_t}\right) + \mu_t$. Using ordinary least squares (OLS), it was found that the elasticity of 2.55% for corporate tax, 1.14% for personal income tax, 1% for indirect tax, 0.74% for security contributions, and 0.35% for government expenditure. Van den Noord (2000) used the method of Giorno et al. (1995) by using the aggregated shares of each in total revenue as weights to derive the elasticity of the total revenue. It was found that the average elasticity of 1.3% for corporate tax, 10.0% for personal income tax, 0.9% for indirect tax, 0.8% for social security, and $-0.3\%$ for current expenditure, and the aggregated CAPB reflected a positive elasticity of 0.49%. The main contribution of Bouthevillain et al. (2001) was based on the analysis of elasticity using the $TB$ tax-based approach share, which is contrary to the traditional approach. Therefore, the elasticity of tax revenue was given by $\epsilon_{tgr} = \sum_{j=1}^{4} TR\left(\frac{TB}{Y}\right)$, and government expenditure was given by $\epsilon_g = G\left(\frac{TB}{Y}\right)$. The time-invariant elasticities of the CAPB on government revenue and expenditure were 1.4% and 0.7%, respectively.

Girouard and André (2006) re-estimated and respecified the elasticity of the CAPB using the OECD framework. Their specification was $\epsilon\_taxw = \left(\sum_{i=1}^{n} \gamma_{iMA_i}\right) / \left(\sum_{i=1}^{n} \gamma_{iAV_i}\right)$, where $\gamma$ denotes the weights of different income distributions, $MA_i$ is the marginal income tax rate, and $VA_i$ is the average income tax rate for each country. It was found that there was an average of $-0.10\%$ sensitivity in the CAPB. Fedelino et al. (2009) used the IMF methodology of constant elasticity concerning the output gap. It was found that government expenditure elasticity was 20.4%, whereas fiscal consolidation negatively impacted demand and growth targets.

Afonso (2010) found that fiscal consolidation harms private consumption. Moreover, budgetary spending categories, including the general government, finally provided support for expansionary fiscal consolidations. Princen et al. (2013) used time-varying parameters to find discretionary tax measures (DTMs). It was found that the average elasticity of the CAPB was $-0.1\%$. Mourre et al. (2014) proposed share tax deviations from the output gap to be used in the CAPB of the OECD, given by $\epsilon_{tgr} = \sum_{j=1}^{4} \frac{TB}{Y}(y - \hat{y})$, and government expenditure was given by $\epsilon_g = \frac{G}{Y}(y - \hat{y})$. It was found that semi-elasticity for revenue was $-0.03\%$, contrary to the positive value of 0.42% obtained using the traditional approach. Dang Price et al. (2014) estimated new tax and expenditure elasticity estimates in the data of Girouard and André (2006). They found that elasticity changes in different tax brackets affect fiscal consolidation episodes. Breuer (2019) adopted the data and methodology of Giorno et al. (1995) and found that the CAPB, which reflects fiscal consolidation results, showed a 0.067% fall in the gross domestic product. Moreover, the authors noted that the CAPB used in the literature has erroneous assumptions that produce flawed results in support of expansionary austerity.

Mourre and Poissonnier (2019) argue that CAPB fiscal semi-elasticities are structural, country-specific, and long-lasting characteristics that are strongly correlated with budgetary variables such as the amount of public spending, spending related to unemployment, and the progress of the tax system. Braz et al. (2019) account for the lag effect in tax and government expenditure data. They found that tax elasticity was 1.07% and the elasticity of direct taxes paid by corporations was 1.95%. They proposed that there is a need for improvements in CAPB output elasticities. Afonso et al. (2022) found that "tax revenue" elasticities have positive Ricardian behaviour, whereby they perceive an increase in taxation to be a sign of future government spending.

No consensus has been reached on the impact of fiscal consolidation on domestic governments. Giavazzi and Pagano (1995), the IMF (2010), Afonso (2010), and Alesina and Ardagna (2010), among others, found that fiscal consolidation of government expenditure reduces government debt and stimulates economic growth. On the other hand, scholars such as Baldacci et al. (2013) and Yang et al. (2015) have shown evidence that fiscal consolidation results in an increase in government debt. Blanchard (1990) outlines that in times of low government debt, fiscal consolidation is successful. Swanepoel and Schoeman (2003) note that when there are high levels of government debt, fiscal consolidation reduces government debt. Müller (2014) argues that fiscal consolidation is self-defeating during financial crises. Monastiriotis (2014) notes that fiscal consolidation leads to unprecedented recessions. Jordà and Taylor (2016) found that a 1% fiscal consolidation translates into a loss of 3.5% of real GDP. Burger and Jimmy (2006) provide evidence that there are two regimes of government debt with a mean of 27.4% and a value of 67% when there is the adoption of fiscal consolidation. Auerbach and Gorodnichenko (2017) note that fiscal consolidation of a government expenditure cut was found to result in a 2.80% fall in government debt. Heimberger (2017) notes that fiscal consolidation has a strong negative association with deep economic crises.

Alesina et al. (2017), in their standard new Keynesian model, demonstrate how sustained expenditure cuts caused by fiscal shocks influence wealth. Under sticky pricing, static distortions brought on by ongoing tax increases result in more significant changes in aggregate supply. Brady and Magazzino (2018) find that in different regimes of high government debt, fiscal consolidation is successful. They found that even when differentiating between different tax types, base broadening during fiscal consolidations resulted in

fewer production and employment reductions than rate increases. Ardanaz et al. (2021) point out that in nations with flexible fiscal rules, the adverse impact of fiscal adjustments on public investment disappears, suggesting that flexible rules shield public investment during episodes of consolidation. The consequence is that assuming productive public investment, the design of fiscal rules can add a growth-friendly dimension to the budgetary sustainability target that has traditionally been the emphasis of fiscal rules. The fiscal rule investigated by Nakatani (2021) used a fiscal reaction function. It was found that natural disasters and climate change affect long-term debt dynamics. The expenditure rule, on the other hand, is based on non-resource and non-grant revenue, which is interdependently defined by budget balance objectives and government debt levels, taking into account projected catastrophe shocks. The implementation of a difference-in-discontinuities was undertaken by Marattin et al. (2022), to investigate revenue- and expenditure-based fiscal consolidation with evidence for the pass-through from federal cuts to local taxes. They note that local governments typically increase taxes as a response to the decline in intergovernmental funding, rather than cutting spending.

## 4. Methodology

This paper uses quantitative analysis to investigate the impact of fiscal consolidation on domestic government debt and measures of the CAPB in South Africa from 1979 to 2022. The theoretical framework of the OECD as well as the IMF extended to the government budget constraint framework, is adopted. The time-varying parameter structural vector autoregression (TVP-VAR) model was used by Primiceri (2005), (Nakajima 2011), and (Koop and Korobilis 2018), among others. There are limitations in this paper in that there are other economic variables that may have not been included in the model. However, Primiceri (2005), (Nakajima 2011), and (Koop and Korobilis 2018) have used a model for monetary policy in this paper for fiscal consolidation analysis. The data were sourced from the South African Reserve Banks (SARB), the IMF, and the Department of the National Treasury Report. The economic variables considered are as follows: $CAPB$ is the cyclically adjusted primary balance, $y_t$ is the potential gross domestic product, $Y_t$ is the gross domestic product, $TGR_t$ is government revenue, $G_t$ is government expenditure, $CAPB\_tgr_t$ is the cyclically adjusted primary balance for government revenue, $CAPB\_g_t$ is the cyclically adjusted primary balance for government expenditure, $rD_t$ is the government debt service payment, $M3_t$ is the money supply proxied, and $GD_t$ is domestic government debt. The framework of the OECD and IMF is shown in Equations (21)–(23).

$$\Delta cnstnt\_elstcy\_\_CAPB\_tgr_t = \sum_{j=1}^{4} TGR_t \left( \frac{y_t}{Y_t} \right)^{\varepsilon_{tgr}} \tag{21}$$

$$\Delta cnstnt\_elstcy\_CAPB\_g_t = G_t \left( \frac{y_t}{Y_t} \right)^{\varepsilon_g} \tag{22}$$

$$\Delta cnstnt\_elstcy\_CAPB_t = \sum_{j=1}^{4} TGR_t \left( \frac{y_t}{Y_t} \right)^{\varepsilon_{tgr}} - G_t \left( \frac{y_t}{Y_t} \right)^{\varepsilon_g} \tag{23}$$

where elasticity is given by $\varepsilon_{tgr}$ and $\varepsilon_g$ is the constant elasticity of government revenue as well as government expenditure. The time-varying elasticity is reflected in Equations (24)–(26).

$$\Delta tvp\_elstcy\_CAPB\_tgr_t = \sum_{j=1}^{4} TGR_t \left( \frac{y_t}{Y_t} \right)^{\varepsilon_{tgr_t}} \tag{24}$$

$$\Delta tvp\_elstcy\_CAPB\_g_t = G_t \left( \frac{y_t}{Y_t} \right)^{\varepsilon_{g_t}} \tag{25}$$

$$\Delta tvp\_elstcy\_CAPB_t = \sum_{j=1}^{4} TGR_t \left( \frac{y_t}{Y_t} \right)^{\varepsilon_{tgr_t}} - G_t \left( \frac{y_t}{Y_t} \right)^{\varepsilon_{g_t}} \tag{26}$$

where $\varepsilon_{tgr} = \varepsilon_{r_t}$ and $\varepsilon_g = \varepsilon_{g_t}$, with the key distinction being the $t$ time subscript reflecting the time-varying elasticity. The theoretical framework is then expanded to the government budget constraint framework to investigate the impact of fiscal consolidation proxied by time-varying CAPB on domestic government debt, as shown in Equations (27) and (28).

$$G_t + rD_t = TGR_t + GD_t + M3_t \tag{27}$$

$$GD_t = G_t + rD_t - TGR_t - M3_t \tag{28}$$

The theoretical framework of the OECD and IMF with time-varying CAPB and the government budget constraint framework is shown in Equations (29)–(31).

$$GD_t = G_t + rD_t - TGR_t - M3_t + tvp\_elstcy\_CAPB\_tgr_t \tag{29}$$

$$GD_t = G_t + rD_t - TGR_t - M3_t + tvp\_elstcy\_CAPB\_g_t \tag{30}$$

$$GD_t = G_t + rD_t - TGR_t - M3_t + tvp\_elstcy\_CAPB_t \tag{31}$$

*Model Specification*

The TVP-VAR model was adopted because it is effective in answering the question of this paper, which is related to finding the time-varying elasticities within the CAPB. TVP-VAR provides time-varying coefficients (Koop and Korobilis 2018) reflecting the responsiveness of the CAPB components that can be attributed to fiscal consolidation. Sims (1980) developed the basic VAR model that was extended by Primiceri (2005), which incorporates time-varying parameters. Nakajima (2011) further improved the framework. The TVP-VAR model is built from the framework of the structural vector autoregressive (SVAR) model, which can then be reduced to the vector autoregressive (VAR) model. The SVAR model is reflected in Equation (32).

$$Ay_t = \beta_0 + \beta_1 y_{t-1} + \beta_1 y_{t-2} + \beta_1 y_{t-3} + \dots \beta_p y_{t-p} + Ce_t \tag{32}$$

where $A$ shows the contemporaneous relationships between the endogenous variables $n * n$ matrix and $p$ shows the number of variables in the system. The subscripts $y_t$, $y_{t-1}$, $y_{t-2}$, and $y_{t-p}$ reflect a matrix $n * 1$ vector of endogenous variables, $\beta_0$ is the intercept, $\beta_1$, $\beta_2$, $\beta_3$, and $\beta_p$ reflect time-invariant coefficients explained by the matrix $n \times n$, $t - p$ indicates the order of autoregression or several lags, and structural shocks in the system are denoted by $E(e_t = 0)$ of the vector that has uncorrelated or orthogonal structural disturbances with a zero mean in the matrix $n \times 1$ (Equation (33)).

$$E\left(e_t,\, e_t'\right) \sum_e = \begin{bmatrix} \sigma_{e_{t1}}^2 & 0 & \cdots & 0 \\ 0 & \sigma_{e_{t2}}^2 & \cdots & \vdots \\ \vdots & \vdots & \ddots & 0 \\ 0 & 0 & \cdots & \sigma_{e_{tn}}^2 \end{bmatrix} \tag{33}$$

where $\sigma$ is the standard deviation, and it is assumed that structural shocks follow a recursive identification pattern with $A$ taking on a lower triangular matrix (Equation (34)).

$$A = \begin{bmatrix} 1 & 0 & \cdots & 0 \\ a_{2,1} & \ddots & \ddots & \vdots \\ \vdots & \ddots & \ddots & 0 \\ a_{n,1} & \cdots & a_{n,p-1} & 1 \end{bmatrix} \tag{34}$$

The SVAR model is transformed through the multiplication of the contemporaneous matrix $A^{-1}$ across all perimeters and is expressed in Equations (35)–(37).

$$A^{-1}Ay_t = A^{-1}\beta_0 + A^{-1}\beta_1 y_{t-1} + A^{-1}\beta_2 y_{t-2} + A^{-1}\beta_3 y_{t-3} + A^{-1}\beta_p y_{t-p} + A_t^{-1}C_{e_t} \quad (35)$$

$$A^{-1}Ay_t = F_0 + A^{-1}F_1 y_{t-1} + A^{-1}F_2 y_{t-2} + A^{-1}F_3 y_{t-3} + A^{-1}F_p y_{t-p} + A^{-1}\sum_e t \quad (36)$$

$$\varepsilon_t \sim (N0, I_n) \quad (37)$$

where $A^{-1}F_i = \beta_1$ for $i = 1 \cdots p$ and $\sum_e t$ is the diagonal matrix denoting the disturbance term. This study uses the rationale of Primiceri (2005) by describing $X_t = I_s \otimes \left(0, y'_{t-1}, y'_{t-2}, \ldots, y'_{t-p}\right)$, $\beta = (F_0, F_1, F_2, F_3 \ldots F_p)$, where $\otimes$ denotes the Kronecker product. The reduced form, VAR, is reflected in Equation (38).

$$y_t = \beta_0 + \beta X_t + A^{-1}\sum_e t \quad (38)$$

The dynamic characteristics of variable interaction and the specification in Equation (38) are further extended to the TVP-VAR model, allowing for the parameters in Equations (39)–(42).

$$y_t = \beta_t X'_t + A_t^{-1}\sum_e t \quad (39)$$

$$\beta_t = \Phi \beta_{t-1} + v_t \quad (40)$$

$$a_t = a_{t-1} + \varsigma_t \quad (41)$$

$$h_t = h_{t-1} + \xi_t \quad (42)$$

where $y_t = X'_{t-1}$ indicates that the variables of interest are explained by the lag function itself, and $\beta_t$, $a_t$, and $h_t$ is indicate the evolution of time-varying parameters following the first-order random walk process, as proposed by Primiceri (2005) and Koop and Korobilis (2018). $\beta_t$ is the time-varying coefficient, $\Phi$ is phi, $a_t$ is the evolution sequence of structural information, and $h_t$ is the evolution sequence of stochastic volatility. On the other hand, $v_t \sim N(0, \Omega_\beta)$, $\varsigma_t \sim N(0, \Omega_a)$, and $\xi_t \sim N(0, \Omega_h)$ denote a new error term note correlated with the matrix shown in Equation (43).

$$V = Var = \begin{bmatrix} t \\ v_t \\ \varsigma_t \\ \xi_t \end{bmatrix} = \begin{bmatrix} I_n & 0 & 0 & 0 \\ 0 & \Omega_\beta & 0 & 0 \\ 0 & 0 & \Omega_a & 0 \\ 0 & 0 & 0 & \Omega_h \end{bmatrix} \quad (43)$$

This paper follows Primiceri (2005) and Koop and Korobilis (2018) by selecting training samples to find the prior information using the ordinary least squares (OLS) algorithm. This information on coefficients factors in the Monte Carlo Markov chain (MCMC) to investigate time-varying parameters. In the MCMC, the Gibbs sampling algorithm is used to fix high-dimensional data. The MCMC discussed above can be expressed in Phases 1–5: Phase 1 has $\beta, a, h, V$, Phase 2 has $\beta|a, h, V, y$; $\Omega_\beta|\beta$, Phase 3 has $a|\beta, h, V, y$; $\Omega_a|a$, Phase 4 has $h|\beta, a, V, y$; $\Omega_h|h$, and Phase 5 returns to Phase 2. One of the variables in this paper is potential GDP, which is used in the CAB. This is calculated using the Hodrick–Prescott filter, as shown in Equation (44).

$$\frac{\text{Min}}{\{Y^p\}_{t=-1}^T} \sum_{t=1}^T (Y_t - Y^p_t)^2 + \lambda \sum_{t=1}^T (Y^p_t - Y^p_{t-1})^2 - (Y^p_t - Y^p_{t-2})^2 \quad (44)$$

## 5. Empirical Results

Table 2 shows the descriptive statistics of economic variables from 1979 to 2022. The *gd* was found to have a mean of 37.22%. The level of *g* was found to have an average of 27.94% between 1979 and 2022. The *tgr* has a growth rate mean of 14.32%. The *m*3 was found to have an index value of 12.79 over the period. The *GDP* was found to be 0.25% between 1979 and 2022 on averageLastly, the economic variable that is considered the $tvp\_elstcy\_CAPB\_tgr$, time-varying elasticity for total government revenue, and $tvp\_elstcy\_g$, time-varying elasticity for government expenditure, were found to have mean values of −0.29% and 0.76%, respectively. The average values are relatively low compared with those found in the case of time-invariant elasticity. For $tvp\_elstcy\_CAPB\_tgr$, $tvp\_elstcy\_CAPB\_g$, and $tvp\_elstcy\_CAPB$, time-varying CAPB was found to have mean values of −1.35%, 5.25%, and 6.61%, respectively.

**Table 2.** Descriptive statistics of the data sourced and estimated.

| Variable | Obs | Mean | Std. Dev. | Min | Max |
| --- | --- | --- | --- | --- | --- |
| *gd* | 44 | 37.22682 | 11.2063 | 21.99 | 73.18 |
| *g* | 44 | 27.94886 | 3.00313 | 23.3 | 37.5 |
| *tgr* | 44 | 14.32779 | 8.75437 | −5.2537 | 36.8419 |
| *m*3 | 44 | 12.795 | 6.12093 | 1.79 | 27.3 |
| *gdp* | 43 | 0.255814 | 2.60926 | −7.7 | 4.4 |
| Estimated data | | | | | |
| $tvp\_elstcy\_CAPB\_tgr$ | 44 | −0.293636 | 12.2292 | −47.3 | 49.72 |
| $tvp\_elstcy\_CAPB\_g$ | 44 | 5.259546 | 45.5113 | −247.4 | 67.85 |
| $tvp\_elstcy\_CAPB$ | 44 | 6.616136 | 56.26286 | −297.13 | 115.15 |

Composed by the authors.

Table A1 shows Dickey–Fuller and Phillips–Perron tests for the unit root with the result that at a level, the unit root null hypothesis could not be rejected, as it was not stationary at the level for all economic variables considered except for *gdp*. All variables were found to be stationary at the first difference. Table A2 shows the optimal length of three lags determined by the standard lag order selection criteria applied to a constant parameter, VAR, which was used in the estimation. The three lag lengths were found using three LR, FPE, and HQIC criteria out of the five criteria selected, while one SBIC criterion selected one and the other AIC criteria selected four.

In Figure 1 Graph (a)–(g) show the Hodrick–Prescott filter for GDP, *dgp* reflects the actual data shown in Graph (a), *dgp_c* is the cyclical component shown in Graph (b), and *dgp_p* reflects the trend component shown in Graph (c). To find a discretionary action that can be attributed to fiscal authorities' action, cyclical movement in economic variables needs to be filtered out (Alesina and Perotti 1995). This is shown in Graphs (e) and (g), and Graphs (d) and (f) show the actual data of government revenue and expenditure, respectively.

Table A3 shows the Johansen tests for cointegration, indicating that there is a long-run relationship between the economic variables and the validity of the use of the VAR model. Table A4 shows the VAR stability condition, reflecting stability in the estimation of the VAR model.

Table 3 shows that the VAR model results in *tgr* reflect a coefficient value of 0.06%, reflecting how responsive the South African fiscal policy authorities are. It was found that the *g* government expenditure output gap share to total output has a negative coefficient of 5.5%. These two elasticities are constant, as reflected in Figure 2 Graph (a) and (d).

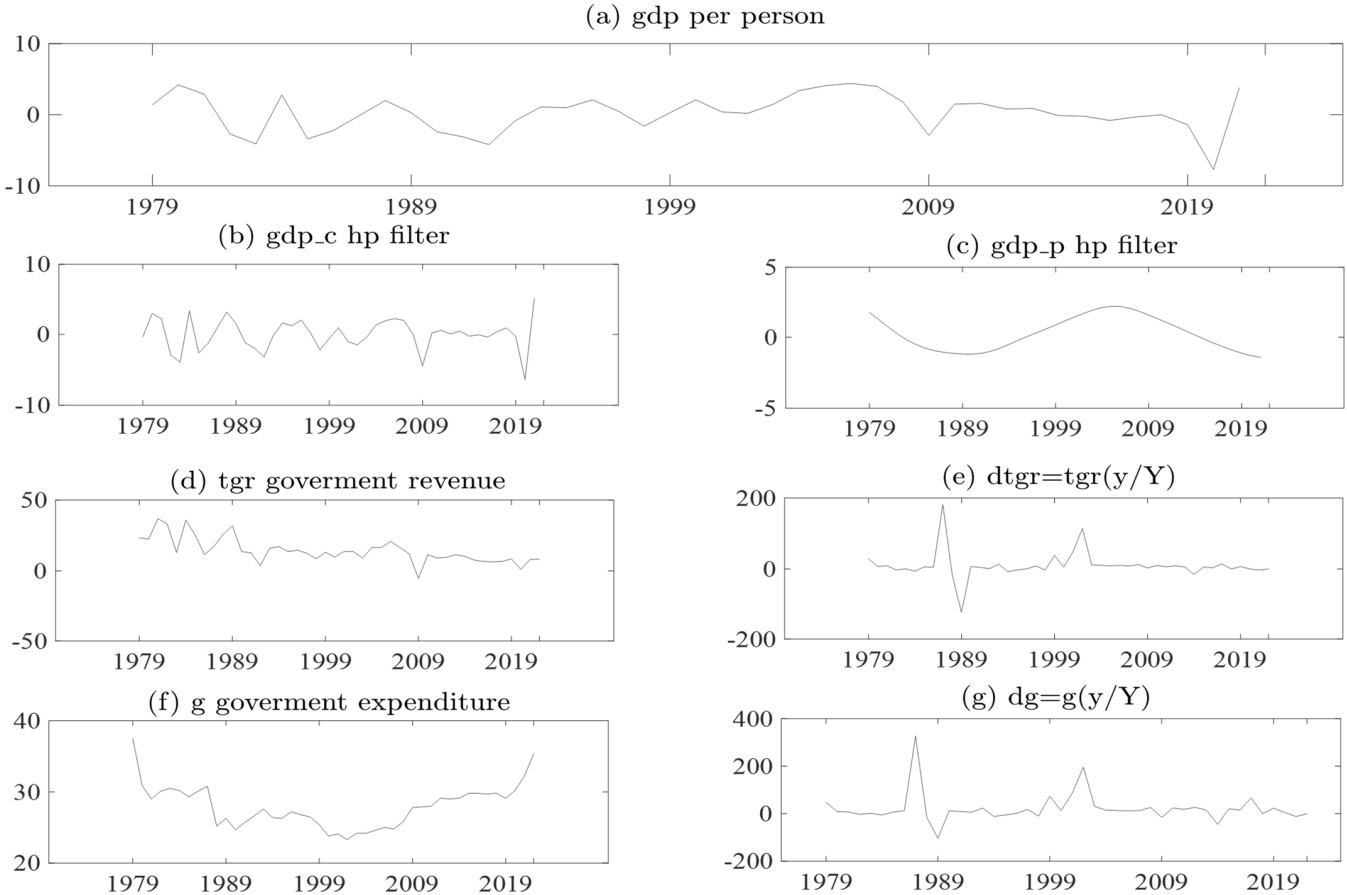

**Figure 1.** Graph (**a**–**g**) is the Hodrick–Prescott filter for GDP. Note Graph (**a**) and *gdp per person* is the gross domestic product, Graph (**b**) and *gdp_c hp filter* is the gross domestic product cyclical component from the Hodrick-Prescott (HP) Filter, Graph (**c**) and *gdp_c hp filter* is the gross domestic product cyclical component from the Hodrick-Prescott (HP) Filter, Graph (**d**) and *tgr* is the total government revenue, Graph (**e**) and *tgr = tgr(y/Y)* is the total government revenue times the proportion of the output gap, Graph (**f**) and *g* is government expenditure, Graph (**g**) and *g = g(y/Y)* is government expenditure times the proportion of the output gap. Composed by the authors.

**Table 3.** VAR model.

| Economic Variables | Estimation 1 | Estimation 2 |
|---|---|---|
| *L3.D.tgr* | 0.330 * (−2.45) | 0.330 * (−2.45) |
| *L2.D.tgr* | 0.0646 ** (−2.65) | 0.0646 ** (−2.65) |
| *L3.D.tgr* | 0.038 (−1.45) | |
| *D.g* | | −5.546 *** (−5.90) |
| *L. ce*1 *cons* | 2.283 (1.02) | 0.000500 (0.00) |
| *N* | 41 | 43 |

*t* statistics in parentheses and * $p < 0.05$, ** $p < 0.01$, *** $p < 0.001$. Composed by the authors.

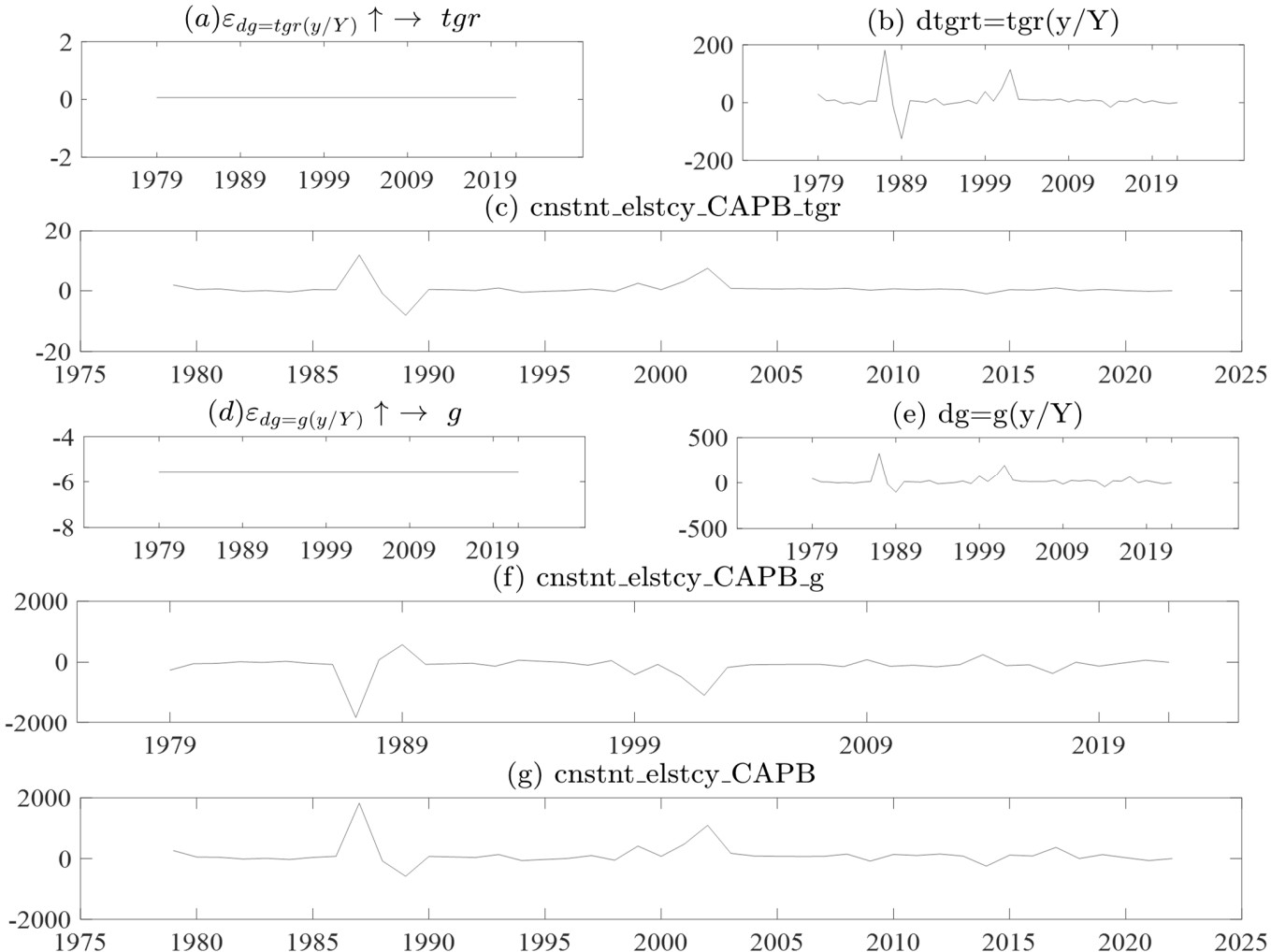

**Figure 2.** Graph (**a**–**g**) is the CAPB with constant elasticity. Note the economic variable that *gdp* is the gross domestic product, *gdp_c hp filter* is the gross domestic product cyclical component from the Hodrick-Prescott (HP) Filter, *gdp_c hp filter* is the gross domestic product cyclical component from the Hodrick-Prescott (HP) Filter, *tgr* is the total government revenue, *tgr = tgr(y/Y)* is the total government revenue times the proportion of the output gap, *g* is government expenditure, and *g = g(y/Y)* is government expenditure times the proportion of the output gap. Composed by the authors.

Figure 2 displays the constant elasticity of the cyclically adjusted primary balance (CAPB) obtained from the estimations presented in Table 2. The CAPB is a measure of a government's fiscal stance that is adjusted to exclude cyclical factors that affect the government's budget balance, such as changes in tax revenues and spending that are due to the economic cycle. The IMF and OECD frameworks use the constant elasticity of the CAPB as a measure of fiscal consolidation, which is the process of reducing a government's budget deficit or debt. Graphs (a) and (d) in Figure 2 show the constant elasticity of total government revenue and expenditure, respectively, which are calculated using the *cnstnt_elstcy_tgr* and *cnstnt_elstcy_g* estimations from Table 2. These elasticities are then multiplied by the output gap, which has been filtered to remove the cyclical component using Equations (21)–(23), resulting in Graphs (b) and (e). Graphs (c) and (f) show the product of the filtered discretionary actions of fiscal authorities *tgr = tgr(y/Y)* and *dg = g(y/Y)* and the corresponding constant elasticities. Finally, Graph (g) displays the time-invariant or constant elasticity of the CAPB. This elasticity represents the responsiveness of the CAPB to changes in the output gap, holding all other factors constant.

The analysis presented in Figure 2 has important policy implications for governments and policymakers. The constant elasticity of the CAPB is a measure of a government's fiscal stance that is adjusted for the effects of the economic cycle, and it provides a useful tool for assessing the effectiveness of fiscal policies in achieving budgetary consolidation. The results from Figure 2 can inform policy decisions by providing information on the appropriate mix of revenue and expenditure measures needed to achieve fiscal consolidation. Policymakers can use the constant elasticities of revenue and expenditure to identify the most effective measures for achieving their desired fiscal targets. For example, if the constant elasticity of revenue is higher than that of expenditure, it may be more effective to focus on revenue-raising measures, such as tax increases, rather than expenditure reductions. Furthermore, the time-invariant elasticity of the CAPB can be used as a benchmark to assess the sustainability of a government's fiscal stance over the long term. If a government's fiscal policy is not aligned with the constant elasticity of the CAPB, it may face challenges in achieving long-term sustainability.

The TVP-VAR results are shown in Tables A4 and A5, which shows the posterior means, standard deviations, 95% credible intervals, convergence diagnostics (CD) of Geweke (1992), and inefficiency factors computed using the MCMC sample. The CD statistics are less than unity, and the inefficiency factors are less than 100. In the estimated result, the null hypothesis of convergence to the posterior distribution is not rejected for the parameters at the 5% significance level based on the CD statistics, and the inefficiency factors are quite low except for $sh2$, which indicates efficient sampling for the parameters and state variables. Figures A1 and A2 shows the sample autocorrelation function, the sample paths, and the posterior densities for the selected parameters. After discarding the initial 2000 samples in the burn-in period, the sample paths appear stable, and the sample autocorrelations drop smoothly. Figures A3 and A4 show the posterior mean estimates for the stochastic volatility of the structural shock used for the estimation of government revenue and expenditure, respectively.

The TVP-VAR results are shown in Tables A5 and A6, which shows the parameters, 95% confidence intervals, convergence diagnostics (CD) of Geweke (1992), and inefficiency factors computed using the MCMC sample. In the estimated result, the null hypothesis of convergence to the posterior distribution is not rejected for the parameters at the 5% significance level based on the CD statistics, and the inefficiency factors are quite low except for sh2, which indicates efficient sampling for the parameters and state variables. Figures A5 and A6 show the time-varying elasticity of government revenue and time-varying elasticity of government expenditure respectively. In both Figures A5 and A6 the Graph of interest if (d), which time-varying elasticity of government revenue and time-varying elasticity of government expenditures noted. Figure A5 Graph (d) the time-varying elasticity of government revenue has been less elastic however the change is seen in the late 1980s to late 1990, thereafter the time-varying elasticity of government revenue inelastic. Figure A6 Graph (d) The time-varying elasticity of government expenditure is reflected with a downwards trend in the from 1790 to late 1990s. Thereafter, time-varying elasticity of government expenditure started to increase.

Figures A7 and A8 show the posterior estimates of stochastic volatility for total government revenue and government expenditure. For both economic variables, the coefficients have 95% credible intervals including the true values. The total government revenue shows again in momentum for volatility from the later 1980s till 2019. On the other hand, government expenditure volatility started to gain momentum of volatility in the early 1990's this was due to most of the government expenditure programs toward democracy in 1994. After 1994 the volatility subsided in 2009 and thereafter start to increase again. Figures A9 and A10 show the evolution sequence of structural information of the interest of total govetment revenue and government expenditure respectily. This will be startting poing of the coeefection in the estimation of the cieffience over time.

Table 4 reflects the descriptive statistics of $tvp\_elstcy\_CAPB$, $CAPB\_IMF$, and $CAB\_IMF$. The $tvp\_elstcy\_CAPB$ has 44 observations, while the data from the IMF have 23 observations.

This reflects that the empirical work of this paper has gone to great lengths to find what can be used to analyse fiscal consolidation. In terms of variation and volatility, the $tvp\_elstcy\_CAPB$ standard deviation is 56.26%. This reflects that there is a 56% variation in the cyclically adjusted primary balance using time-varying elasticity. The difference in the number of observations between the tvp_elstcy_CAPB variable and the IMF data suggests that different methods were used to generate these variables. This could have implications for the reliability and comparability of the data. The difference in the number of observations between the $tvp\_elstcy\_CAPB$ variable and the IMF data could have important implications for the reliability and comparability of the data. This means that policymakers should be cautious when interpreting data from different sources and should take steps to ensure that the data they are using is as accurate and reliable as possible.

**Table 4.** Estimated CAPB for this paper and the IMF.

| Variable | Obs | Mean | Std. Dev. | Min | Max |
|---|---|---|---|---|---|
| IMF data of CAPB for South Africa | | | | | |
| $tvp\_elstcy\_CAPB$ | 44 | 6.616136 | 56.26286 | −297.13 | 115.15 |
| IMF data of CAPB for South Africa | | | | | |
| $CAPB\_IMF$ | 23 | 0.3516041 | 2.361112 | −4.687403 | 3.766418 |
| $CAB\_IMF$ | 23 | −3.063716 | 2.552726 | −9.054413 | 0.8080437 |

Composed by the authors.

Figure 3 Graph (a)–(g) shows the CAPB with time-varying constant elasticity. Graphs (a) and (d), show the time-varying elasticity of fiscal consolidation, which is in contrast to the constant elasticity applied in the IMF and OECD frameworks. To represent time-varying elasticity in fiscal consolidation for $tgr$ government revenue, the data in Graphs (a) and (b) are multiplied to obtain $tvp\_elstcy\_CAPB\_tgr$, which reflects the CAPB with time-varying elasticity shown in Graph (c). The time-varying elasticity reflected in Graph (d) is for $g$, government expenditure. The time-varying elasticity for $g$ (government expenditure) multiplied by the share of GDP deviation is shown in Graph (e), resulting in the $tvp\_elstcy\_CAPB\_g$ time-varying CAPB for government expenditure. The $tvp\_elstcy\_CAPB$ has a range from −297.13% as the minimum value to 115.5% as the maximum value. In the first 8 years, the $tvp\_elstcy\_CAPB$ was characterized by an average value of 26.36%. In the same period, the maximum value found was 45.85% in 1989, and the lowest value found was –11.06% in 1985. The $tvp\_elstcy\_CAPB$ drastically fell to −297.13%, which is thought to be an outlier in the data. Observations of the data reflect that this could be related to the fall in the gross domestic product per person in the same year. The $tvp\_elstcy\_CAPB$, the next point, was found to be high in 1989, which recorded a value of 115.15%. After 1989, $tvp\_elstcy\_CAPB$ started to become stable in terms of volatility. Between 1989 and 2022, the $tvp\_elstcy\_CAPB$ recorded an average of 6.76% as well as a maximum value of 75.30% and a minimum value of −91.40%.

The comparison between the time-varying constant elasticity approach and the constant elasticity approach used in the IMF and OECD frameworks underscores the importance of considering the complexity of fiscal consolidation measures and the potential impacts of these measures over time. By using a more nuanced approach, policymakers can gain a better understanding of the potential benefits and drawbacks of different fiscal consolidation measures and can design policies that are more effective and sustainable over the long term. The range of values in the tvp_elstcy_CAPB variable, which includes both very high positive and negative values, reflects the high volatility and complexity of fiscal consolidation measures. The outcome sheds light on how the cyclically adjusted primary balance changes over time and emphasizes the value of employing time-varying elasticity methods to capture fiscal consolidation.

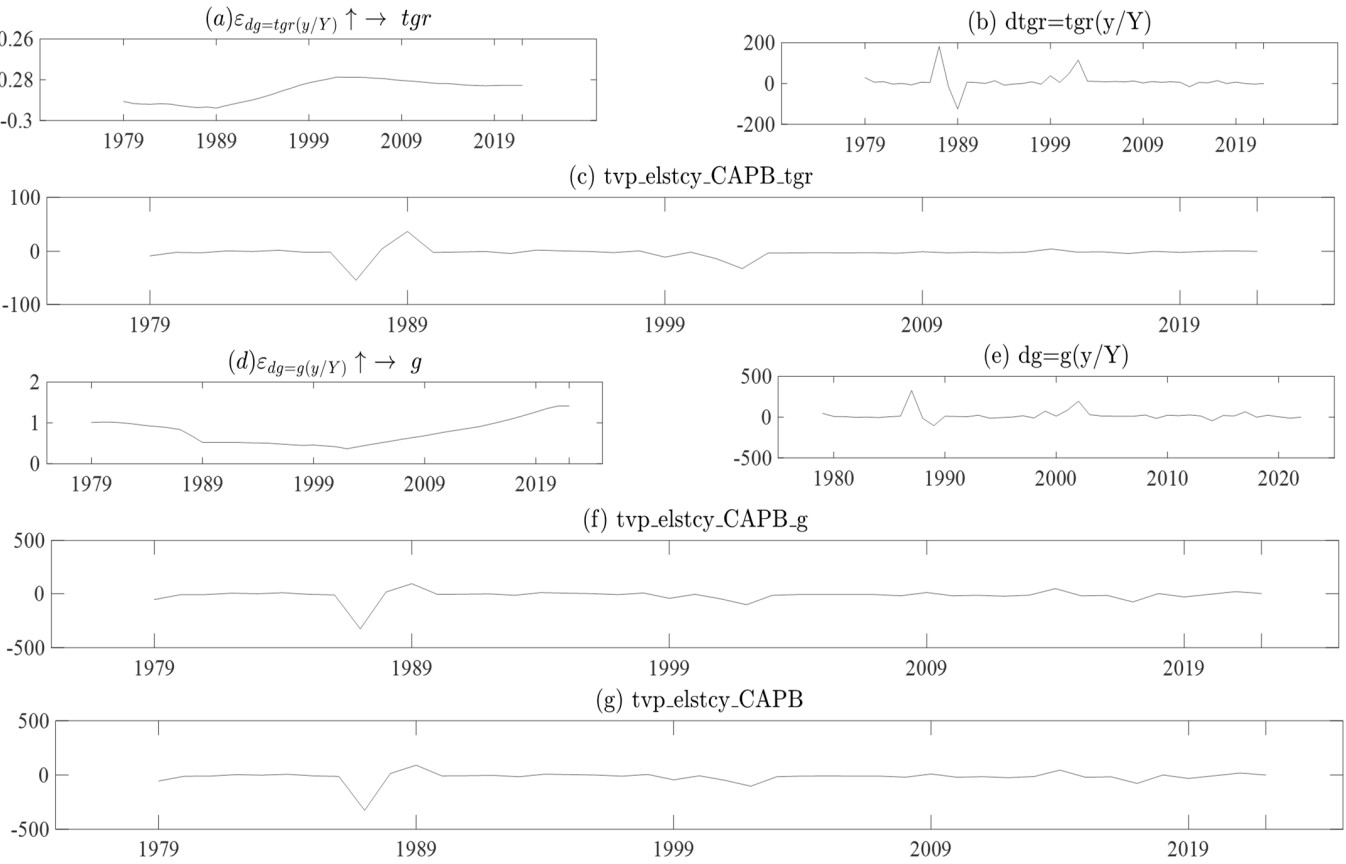

**Figure 3.** Graph (**a**–**g**) is the CAPB with time-varying constant elasticity. Where Graph (**a**,**d**) and $dg = g(y/Y)$ shows the government expenditure times the proportion of potation gross domestic product, Graph (**b**,**e**) and $tgr = g(y/Y)$ shows the g total government revenue times the proportion if potation gross domestic product, Graph (**c**) and $tvp\_elstcy\_CAPB\_tgr_t$ is the time-varying cyclically adjusted primary balance for government revenue, Graph (**f**) and $tvp\_elstcy\_CAPB\_g_t$ is the time-varying elasticity cyclically adjusted primary balance for government expenditure and Graph (**g**) and $tvp\_elstcy\_CAPB_t$ is the cyclically adjusted primary balance for government expenditure. Composed by the authors.

### 5.1. Comparison of the CAPB of This Paper and the IMF

The introduction of the cyclically adjusted primary balance calculation by the IMF in 2000 was a significant development in fiscal policy analysis. This measure is used to assess the sustainability of fiscal policy by removing the cyclical component of government revenues and expenditures and thus providing a clearer picture of the underlying structural fiscal position of a country.

Figure 4 Graphs (a) and (b) show the $tvp\_elstcy\_CAPB\_g$ with time-varying elasticity, which provides a more dynamic approach to analyzing the cyclically adjusted primary balance. The time-varying elasticity captures changes in the responsiveness of the primary balance to the output gap over time and is calculated using a state-space model. On the other hand, Graphs (c) and (d) show the cyclically adjusted primary balance using time-invariant elasticity for the methodology of the IMF, which is based on a fixed elasticity of government revenue and expenditure with respect to the output gap. It is worth noting that the IMF did not calculate the cyclically adjusted primary balance before 2000, as seen in Graphs (c) and (d). Therefore, the introduction of this measure by the IMF in 2000 was a significant step in providing a more comprehensive assessment of the fiscal sustainability of member countries. Both $CAPB\_IMF$ and $CAB\_IMF$ reflect downward information trends from 2000 to 2020. These data reflect that South Africa has been adopting a less discretionary fiscal policy or fiscal consolidation to reduce domestic government debt.

However, with the data that were calculated in this paper using time-varying elasticity from 2000 to 2020, the CAPB has a positive trend. Contrary to the IMF, this trend reflects that there has been a discretionary fiscal policy or fiscal consolidation in the effort to reduce domestic government debt.

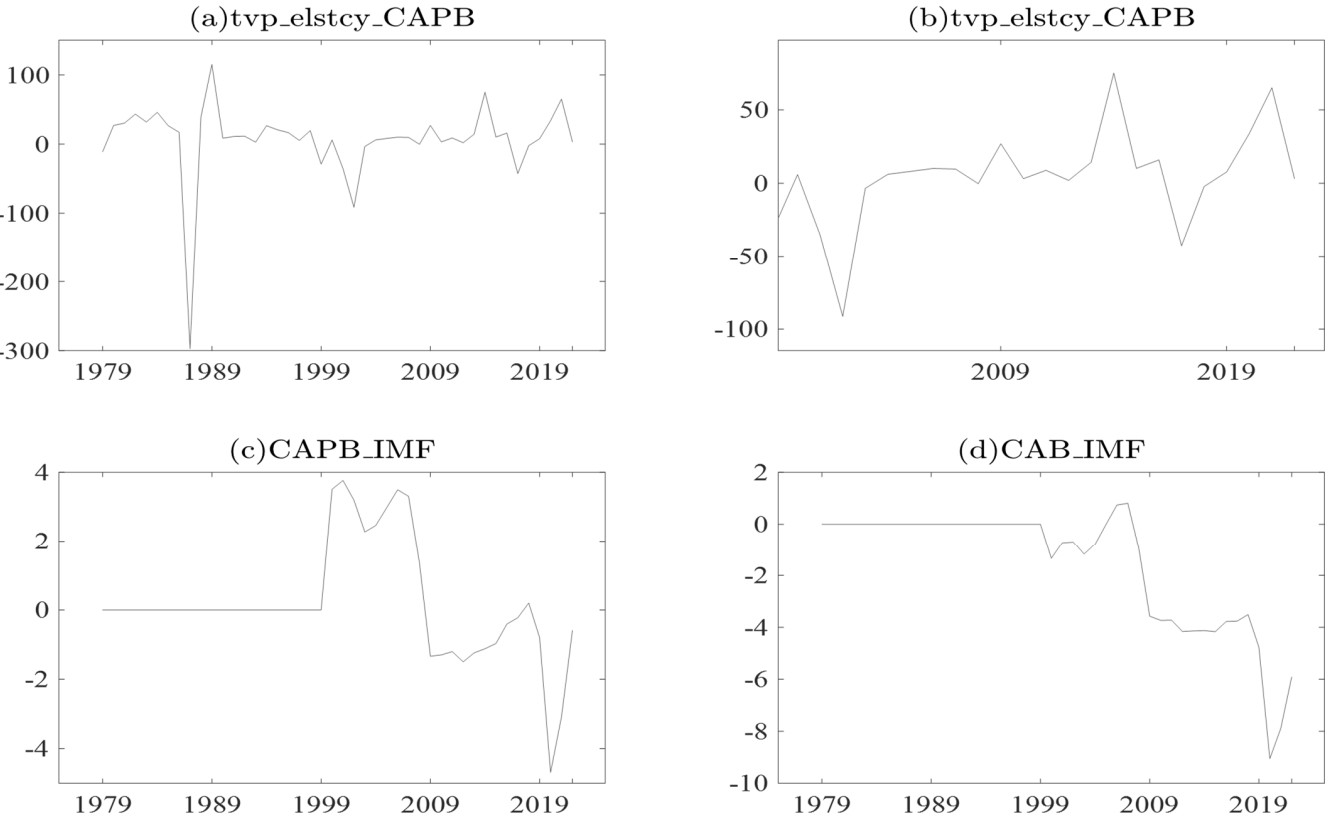

**Figure 4.** Graph (**a–d**) show the time-varying CAPB of this paper and time-invariant CAPB of the IMF. Where Graph (**a**,**b**) *tvp_elstcy_CAPB* show the cyclically adjusted primary balance with time-varying elasticity. While Graph (**c**,**d**), as well as *CAPB_IMF* shows the cyclically adjusted primary balance from the International Monetary Fund. Composed by the authors.

*5.2. Impact of Fiscal Consolidation of Domestic Government Debt Using Time-Varying CAPB*

Figure 5 shows a time-varying CAPB that proxies fiscal consolidation impact in the economic variables of interest. The $\varepsilon_{tvp\_elstcy\_CAPB\_tgr} \uparrow \rightarrow gd$ shock of fiscal consolidation through taxes or government revenue is anticipated to be implemented in 3 years, with the highest multiplier being −0.5% in Year 1. This results in a fall in *gd* in the following year.

After Year 1, *gd* starts to become unstable and increases above equilibrium. However, fiscal consolidation is anticipated to be implemented in 6 years, with a multiplier of 0.9%. The fiscal consolidation expected in 12 years has the highest multiplier at 0.5%. Fiscal consolidation has a detrimental effect on *gd*. On the other hand, the $\varepsilon_{tvp\_elstcy\_CAPB\_g} \uparrow \rightarrow gd$ shock of fiscal consolidation through government expenditure expected in 3 years increases *gd* in Year 1; after that year, there is a reduction in *gd* until Year 3, with a maximum multiplier value of 0.05%. After Year 3, *gd* increases and returns to equilibrium. The shock $\varepsilon_{tvp\_elstcy\_CAPB\_g} \uparrow \rightarrow gd$ expected in 6 years is found to reflect high volatility in *gd*. First, there is a drastic reduction in *gd* in Year 1 followed by a high increase in the following year and a high multiplier value of 0.025%. In Year 1, the results are similar to those of Giavazzi and Pagano (1995), the IMF (2010), and Afonso (2010), outlining that a government expenditure cut results in a reduction in *gd*. The shock of $\varepsilon_{tvp\_elstcy\_CAPB\_g} \uparrow \rightarrow gd$ expected in 12 years increases *gd* in the 1st year, with a maximum multiplier value of −0.022%. Thereafter, there is a reduction in *gd* at a level below equilibrium from Year 2 to Year 11, and in Year 12, it returns to equilibrium.

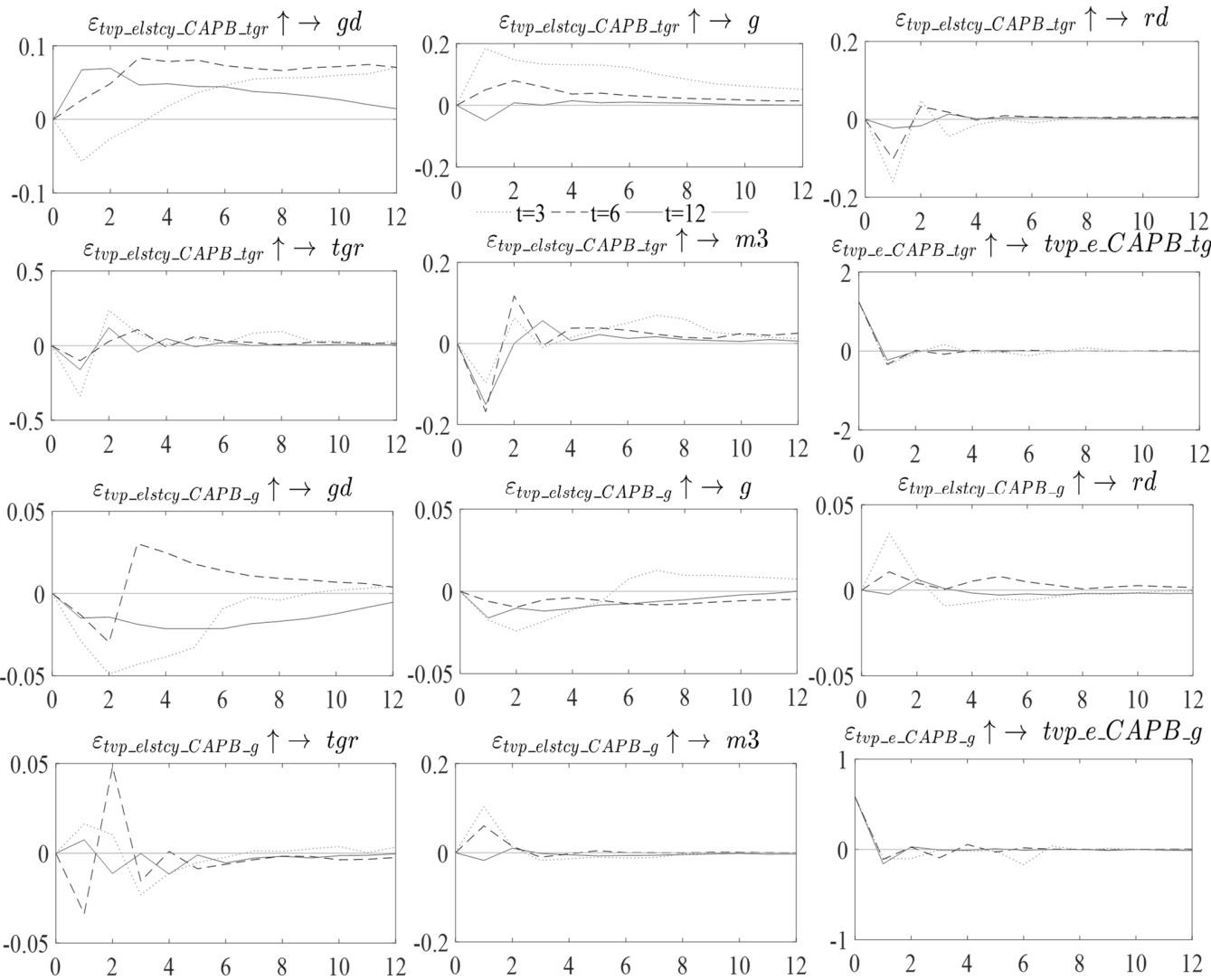

**Figure 5.** Time-varying impulse response functions. Where $gd_t$ is the doedomesticvernment debt, $g_t$ is government expenditure, $rd_t$ is the government debt service payment, $tgr_t$ is government revenue, $m3_t$ is the money supply, and $CAPB\_g_t$ is the cyclically adjusted primary balance for government expenditure. Composed by the authors.

## 6. Conclusions

This paper is based on the investigation of new measures of fiscal consolidation episodes with the use of the cyclical adjusted primary balance (CAPB) taking into account the time-varying elasticity. This investigation is based on the problem that is identified that IMF and OECD used constant elasticity when calculating the CAPB and not time-varying elasticity. Moreover, another broad question of the paper is to investigate what is the impact of fiscal consolidation on domestic government debt. The empirical work uses data from 1979 to 2020 and the Time-Varying Parameter Structural Vector Auto-Regression (TVP-VAR model) (Nakajima 2011) to find time-varying elasticity. The economic variables utilized in this empirical work are domestic government debt, money supply, total government revenue, debt service ratio, and fiscal consolidation proxied by CAPB. The empirical work found that IMF data for the CAPB ran over 23 years, but the one that is calculated in this study is 43 years. The IMF reflects that the CAPB has no upward trend, indicating that fiscal authorities have been taking less and less discretionary action toward stabilizing the economy. However, when the time-varying elasticity is accounted for, it is found that there is variation in the CAPB reflecting different times of discretionary action by the fiscal

authority in the effort to stabilize the economy. There is a 56.26% variation in the CAPB with time-varying elasticity and there is a 2.36% variation in the CAPB of the IMF data.

Regarding the investigation of the impact of fiscal consolidation on domestic government debt, the time-varying cyclical adjusted primary balance is used to proxy fiscal consolidation. In the TVP-VAR model, it is found that fiscal consolidation is expected to occur in 3 years and 6 years resulting in a radical increase in domestic government debt in the first 3 years of implementation. Moreover, the domestic government debt remains at a high level from year 3 until year 12. If fiscal consolidation is expected in 12 years, this will result in a fall in the domestic government debt in the first 3 years when the fiscal consolidation has been implemented. However, after that, the domestic government debt will increase up until year 12. Given that the South Africa budgetary planning is done over 3 years, the empirical work of this study concludes that fiscal consolidation years will increase domestic government debt in South Africa. Moreover, it is recommended that the fiscal consolidation be planned 3 years to 6 years in advance. Nevertheless, fiscal authorities need to critically examine the benefit of fiscal consolidation in the short run. Future studies need to also investigate the long-run implication of fiscal consolidation.

The policy implication based on the finds of this paper is the value of employing time-varying elasticity when calculating the cyclically adjusted primary balance (CAPB) to provide a more precise evaluation of the independent steps taken by fiscal authorities to stabilize the economy. Because of the large short-term effects of fiscal consolidation on domestic government debt, careful planning is required, as is taking into account any long-term negative effects on government debt levels. Constant elasticity has drawbacks that can lead to an erroneous picture of the fiscal authorities' discretionary decisions when used to calculate the CAPB. Further research is required to evaluate the long-term effects of fiscal consolidation and its effects on other economic factors outside government debt levels. On the other hand, it is recommended that when computing the CAPB, the IMF and OECD should use time-varying elasticity to more accurately reflect the discretionary measures taken by fiscal authorities. The short-term advantages of fiscal consolidation should be carefully considered, and fiscal authorities should prepare for any potential negative effects on domestic government debt. To ensure adequate planning and prevent any negative effects on the level of government debt, budgetary planning should be done over a longer period. While deciding on a policy, fiscal authorities should take into account how fiscal consolidation may affect other economic factors outside government debt levels, such as economic growth and unemployment.

**Author Contributions:** Conceptualization, E.M.B.; methodology, E.M.B.; software, E.M.B.; validation, E.M.B.; formal analysis, E.M.B., and P.N.; investigation, E.M.B.; resources, E.M.B. and P.N.; data curation, E.M.B.; writing—original draft preparation, E.M.B.; writing—review and editing, E.M.B. and P.N.; visualization, E.M.B.; supervision, P.N.; project administration, E.M.B.; funding acquisition, E.M.B., and P.N. All authors have read and agreed to the published version of the manuscript.

**Funding:** This research was funded by University of KwaZulu-Natal grant number 2023-1-210542387 And The APC was funded by University of KwaZulu-Natal.

**Institutional Review Board Statement:** Not applicable.

**Informed Consent Statement:** Not applicable.

**Data Availability Statement:** The data is available on request.

**Acknowledgments:** My immense gratitude goes P. Nyantanga co-author and supervisor as well as Ntokozo Nzimande for giving material to read on time-varying models.

**Conflicts of Interest:** The authors declare no conflict of interest.

## Appendix A

**Table A1.** Dickey–Fuller and Phillips–Perron tests for unit root.

| Variables | | Dickey-Fuller Test for Unit Root | | | | Phillips–Perron Test for Unit Root | | | |
|---|---|---|---|---|---|---|---|---|---|
| | | **Test** | **1%** | **5%** | **10%** | **Test** | **1%** | **5%** | **10%** |
| *d.gd* | Z(t) | −3.902 | −3.634 | −2.952 | −2.61 | −3.924 | −3.634 | −2.952 | −2.61 |
| *d.g* | Z(t) | −7.018 | −3.634 | −2.952 | −2.61 | −7.124 | −3.634 | −2.952 | −2.61 |
| *d.tgr* | Z(t) | −9.221 | −3.634 | −2.952 | −2.61 | −3.506 | −3.628 | −2.95 | −2.608 |
| *d.m3* | Z(t) | −6.166 | −3.634 | −2.952 | −2.61 | −2.677 | −3.628 | −2.95 | −2.608 |
| *gdp* | Z(t) | −4.71 | −3.634 | −2.952 | −2.61 | −4.649 | −3.634 | −2.952 | −2.61 |
| *d.rd* | Z(t) | −9.403 | −3.736 | −2.994 | −2.628 | −10.045 | −3.736 | −2.994 | −2.628 |

MacKinnon's approximate *p*-value for Z(t) = 0.0000. The number of obs = 42. Composed by the authors.

**Table A2.** Selection-order criteria.

| | | | | | Selection-Order Criteria for Variables | | | |
|---|---|---|---|---|---|---|---|---|
| **Lag** | **LL** | **LR** | **df** | **p** | **FPE** | **AIC** | **HQIC** | **SBIC** |
| 0 | −340.914 | | | | 95795 | 17.1457 | 17.1762 | 17.2301 * |
| 1 | −336.294 | 9.2392 | 4 | 0.055 | 92,923.1 | 17.1147 | 17.2063 | 17.368 |
| 2 | −330.807 | 10.974 | 4 | 0.027 | 86,443.4 | 17.0404 | 17.193 | 17.4626 |
| 3 | −324.851 | 11.913 * | 4 | 0.018 | 78,752.5 * | 16.9425 | 17.1563 * | 17.5336 |
| 4 | −320.78 | 8.1412 | 4 | 0.087 | 79,135.9 | 16.939 * | 17.2138 | 17.699 |

Sample: 1983–2022, number of obs = 43. * denotes rejection of the hypothesis at the 0.05 level. Composed by the authors.

**Table A3.** Johansen test for cointegration.

| Maximum Rank | Parms | LL | Eigenvalue | Trace Statistic | 5% Critical Value |
|---|---|---|---|---|---|
| 0 | 6 | −368.913 | | 36.9264 | 15.41 |
| 1 | 9 | −353.482 | 0.52042 | 6.0628 | 3.76 |
| 2 | 10 | −350.45 | 0.13442 | | |

Trend: constant, number of obs = 42, sample: 1981–2022, lags = 3. Max eigenvalue test indicates 0 cointegrating equation(s) at the 0.05 level. Composed by the authors.

**Table A4.** VAR stability condition.

| | Eigenvalue Stability Condition | |
|---|---|---|
| **Eigenvalue** | **Eigenvalue** | **Modulus** |
| 0.8781253 | | 0.87813 |
| −0.391861 | +0.6125297i | 0.72715 |
| −0.391861 | −0.6125297i | 0.72715 |
| 0.0827303 | +0.5868185i | 0.59262 |
| 0.0827303 | −0.5868185i | 0.59262 |
| 0.0545967 | | 0.0546 |

Composed by the authors.

**Table A5.** Estimated parameters in the TVP-VAR model for government expenditure.

| Parameter | Mean | Stdev | 95% U | 95% L | Geweke | Inef. |
|---|---|---|---|---|---|---|
| sb1 | 0.0029 | 0.0006 | 0.002 | 0.0043 | 0.157 | 9.97 |
| sb2 | 0.0028 | 0.0006 | 0.002 | 0.0042 | 0.346 | 6.59 |
| sa1 | 0.0056 | 0.0016 | 0.0034 | 0.0097 | 0.912 | 12.51 |
| sa2 | 0.0058 | 0.0022 | 0.0034 | 0.0109 | 0.319 | 26.66 |
| sh1 | 0.0056 | 0.0017 | 0.0034 | 0.01 | 0.995 | 14.81 |
| sh2 | 1.5984 | 0.4177 | 0.9125 | 2.5323 | 0.42 | 11.63 |

TVP-VAR model (Lag = 1), Iteration: 20,000 and Composed by the authors.

**Table A6.** Estimated parameters in the TVP-VAR model total government revenue.

| Parameter | Mean | Stdev | 95% U | 95% L | Geweke | Inef. |
|---|---|---|---|---|---|---|
| sb1 | 0.1937 | 0.1175 | 0.0298 | 0.4728 | 0.409 | 206.57 |
| sb2 | 0.5593 | 0.2142 | 0.202 | 1.0391 | 0.841 | 148.62 |
| sa1 | 0.0055 | 0.0017 | 0.0034 | 0.0096 | 0.119 | 9.93 |
| sa2 | 0.0024 | 0.0003 | 0.0019 | 0.0031 | 0.12 | 0.54 |
| sh1 | 0.0024 | 0.0003 | 0.0019 | 0.0031 | 0.762 | 1.11 |
| sh2 | 0.1937 | 0.1175 | 0.0298 | 0.4728 | 0.409 | 206.57 |

TVP-VAR model (Lag = 1), Iteration: 20,000 and Composed by the authors.

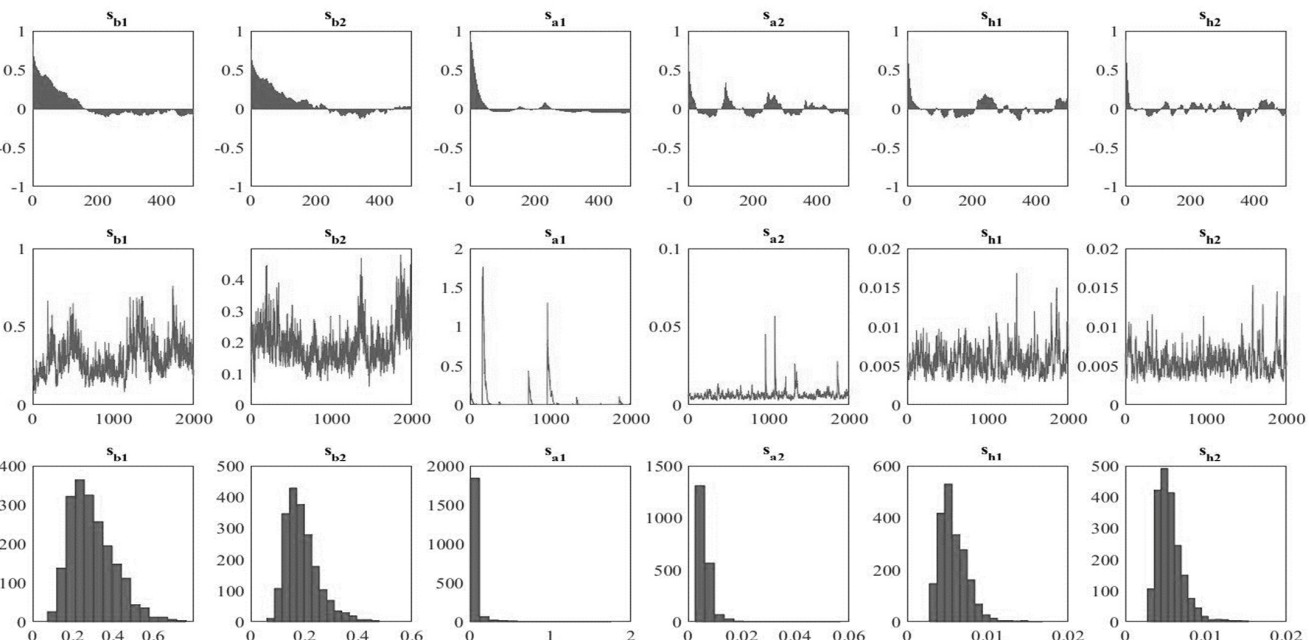

**Figure A1.** Estimates of the moments and posterior distributions of the model for *G*. Note: The estimates of $\Sigma_a$ and $\Sigma_\beta$ are multiplied by 100. Composed by the authors.

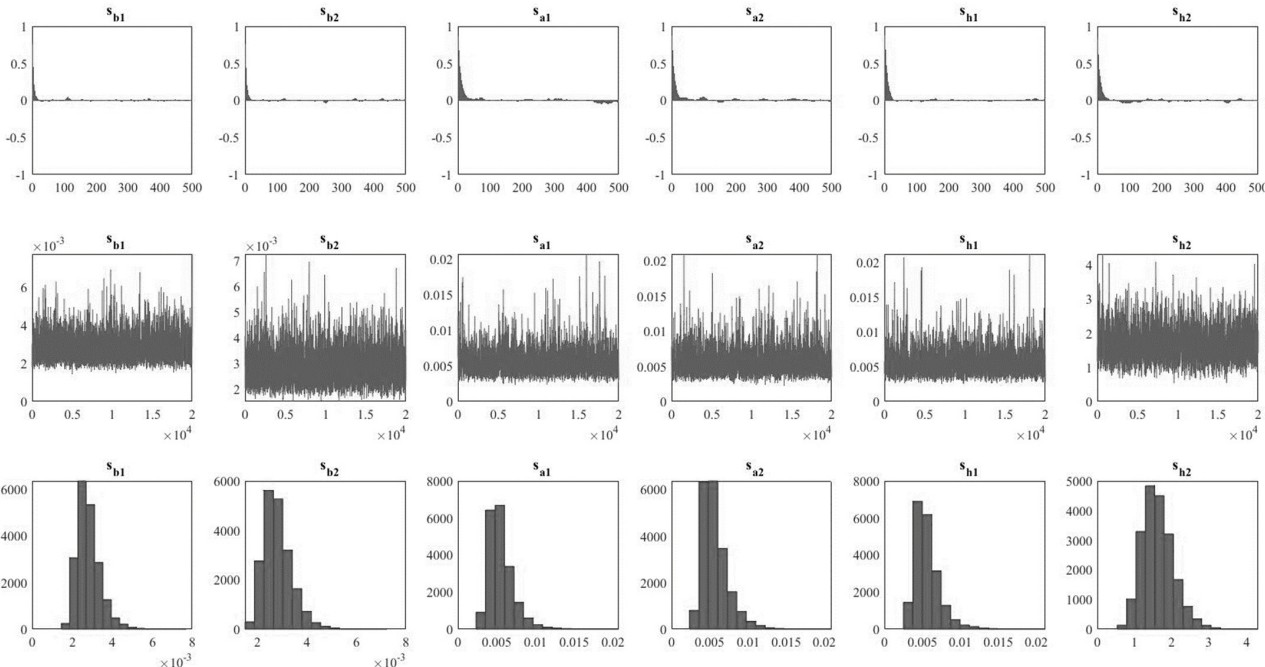

**Figure A2.** Estimates of the moments and posterior distributions of the model for *TGR*. Note: The estimates of $\Sigma_a$ and $\Sigma_\beta$ are multiplied by 100. Composed by the authors.

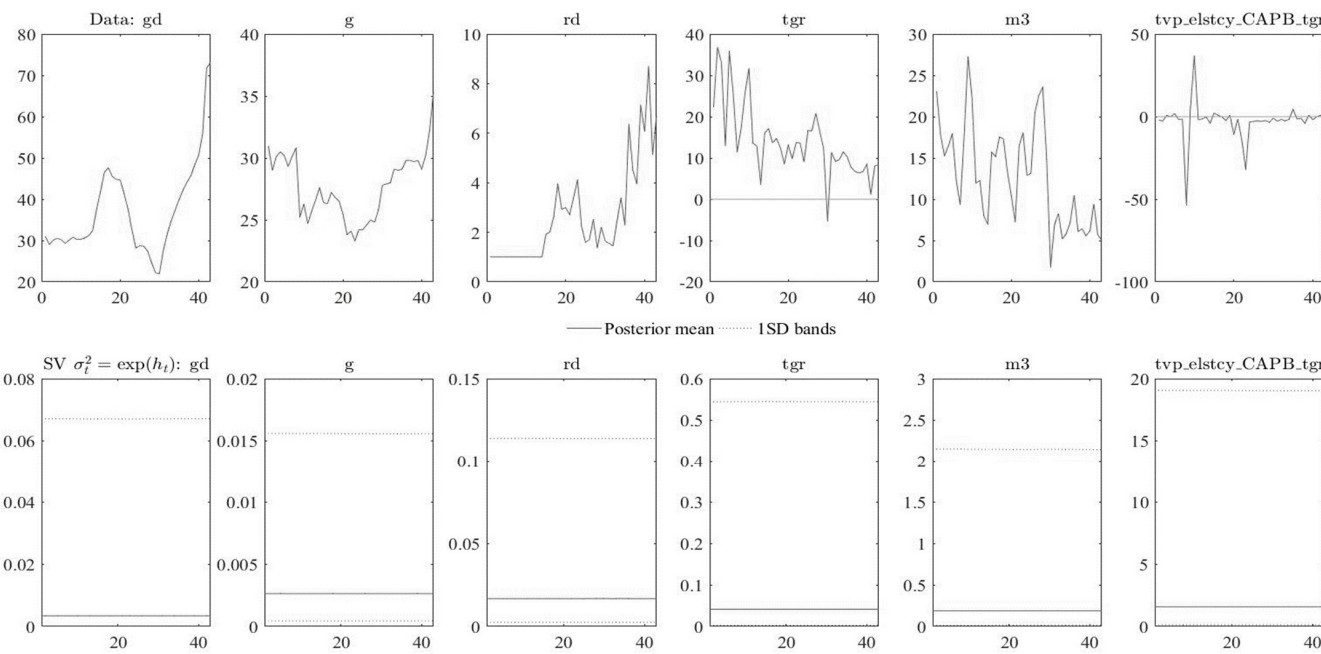

**Figure A3.** Posterior mean estimates for stochastic volatility of the structural shock for *TGR*. Where $gd_t$ is domestic government debt, $g_t$ is government expenditure, $rd_t$ is the government debt service payment, $tgr_t$ is government revenue, $m3_t$ is the money supply, and $CAPB\_tgr_t$ is the cyclically adjusted primary balance for government revenue. Composed by the authors.

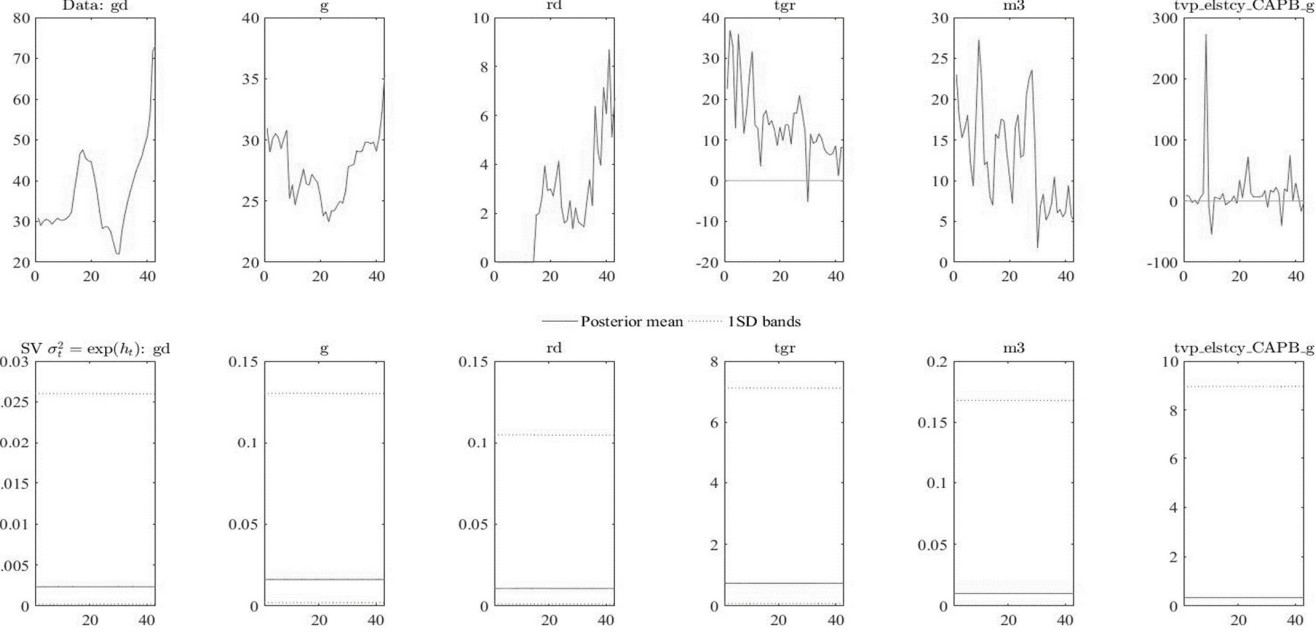

**Figure A4.** Posterior mean estimates for stochastic volatility of the structural shock for *G*. Where $gd_t$ is domestic government debt, $g_t$ is government expenditure, $rd_t$ is the government debt service payment, $tgr_t$ is government revenue, $m3_t$ is the money supply, and $CAPB\_g_t$ is the cyclically adjusted primary balance for government expenditure. Composed by the authors.

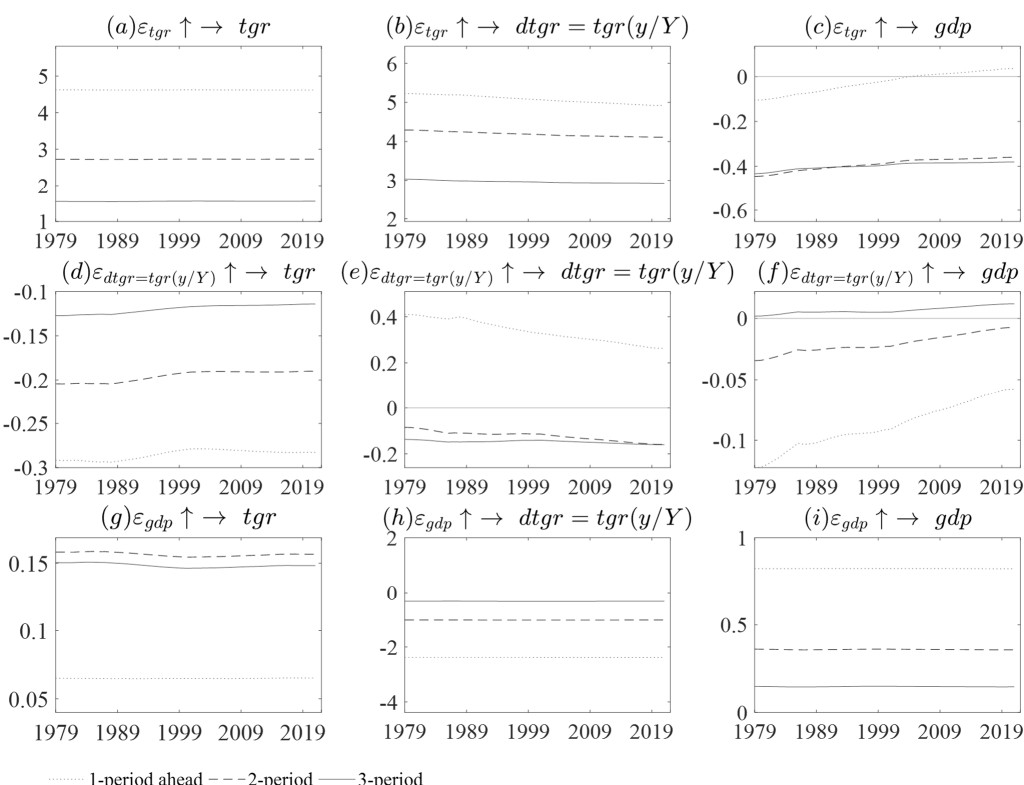

**Figure A5.** Graph (**a–i**) is the time-varying elasticity of government revenue. Where *tgr* is the total government revenue $tgr = g(y/Y)$ is the g total government revenue times the proportion of potation gross domestic product and *gdp* is the gross domestic product per person. Composed by the authors.

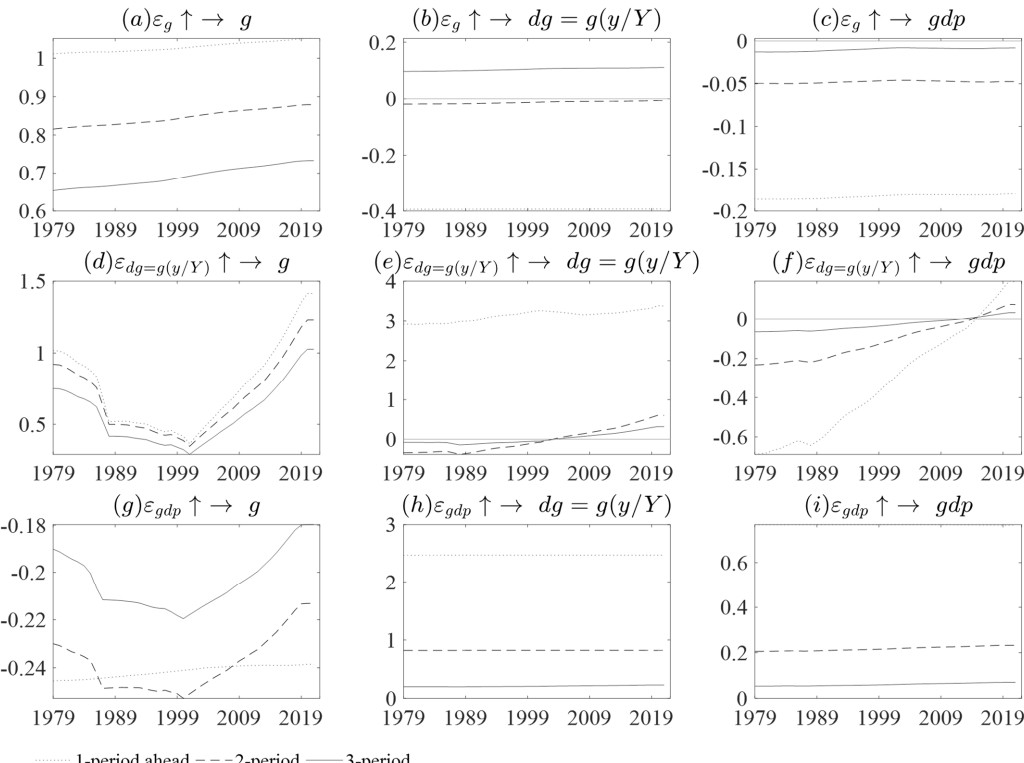

**Figure A6.** Graph (**a–i**) is the time-varying elasticity of government expenditure. Where *g* is the government expenditure $dg = g(y/Y)$ is the government expenditure times the proportion of potation gross domestic product and *gdp* is the gross domestic product per person. Composed by the authors.

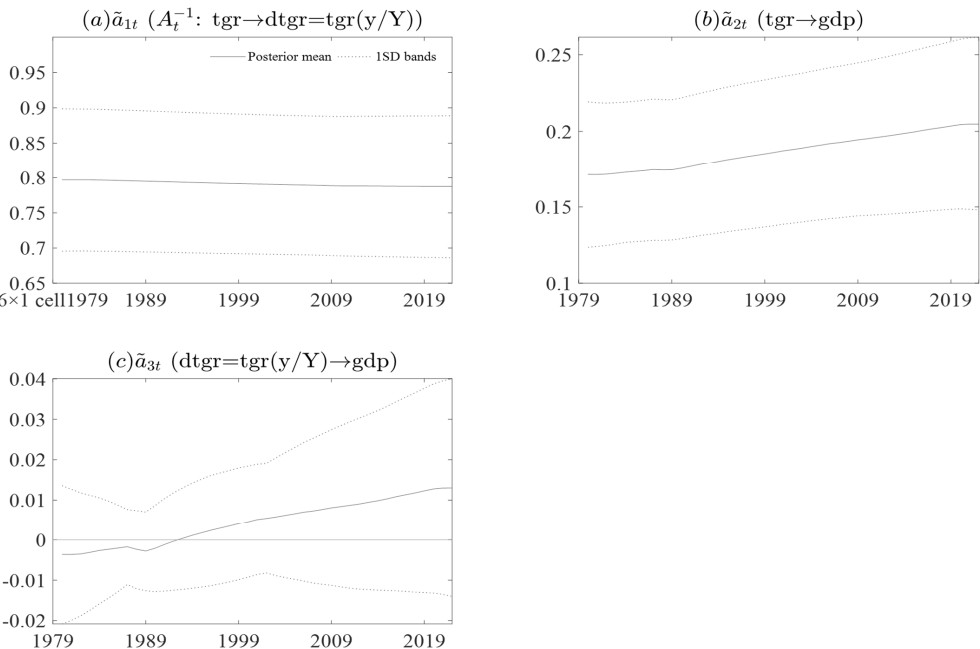

**Figure A7.** Graph (**a–c**) is the posterior draws for each data series for government revenue. Estimation results of $\bar{a}_{1,t}$ on the TVP regression model for the simulated data. True value (solid line), posterior mean (bold) and 95% credible intervals (dashed). The True models are Markov-switching coefficient and stochastic volatility. The TVP regression model with time-varying coefficient and stochastic volatility is fitted. Where $tgr$ is the total government revenue $tgr = g(y/Y)$ is the g total government revenue times the proportion of potation gross domestic product and $gdp$ is the gross domestic product per person. Composed by the authors.

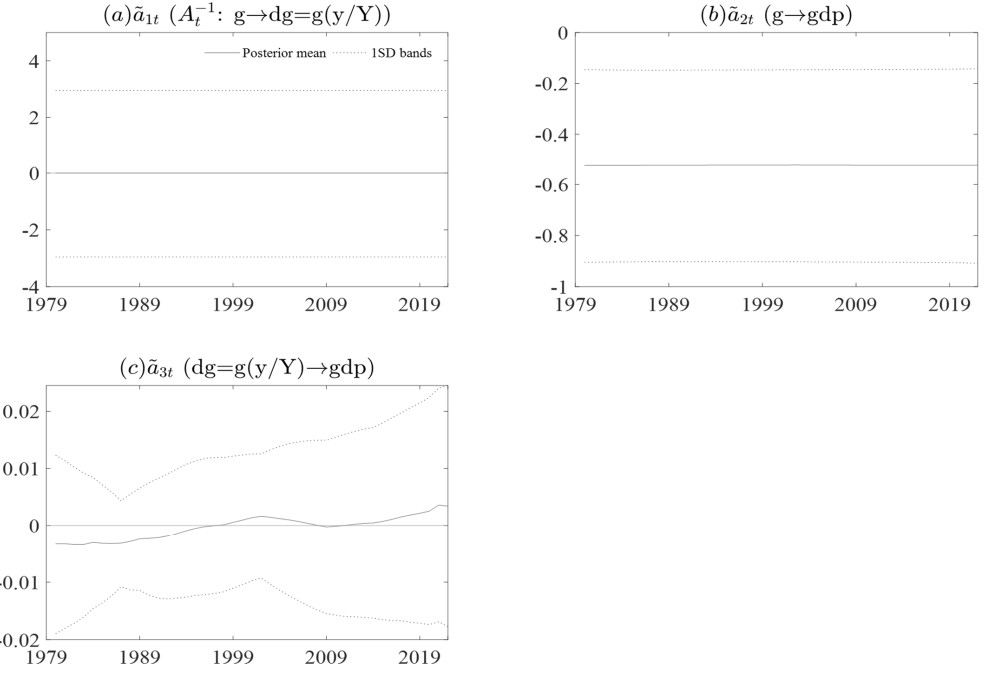

**Figure A8.** Graph (**a–c**) is the posterior draws for each data series for government expenditure. Estimation results of $\bar{a}_{1,t}$ on the TVP regression model for the simulated data. True value (solid line), posterior mean (bold) and 95% credible intervals (dashed). The True models are Markov-switching coefficient and stochastic volatility. The TVP regression model with time-varying coefficient and stochastic volatility is fitted. Where $g$ is the government expenditure $dg = g(y/Y)$ is the government expenditure times the proportion of potation gross domestic product and $gdp$ is the gross domestic product per person. Composed by the authors.

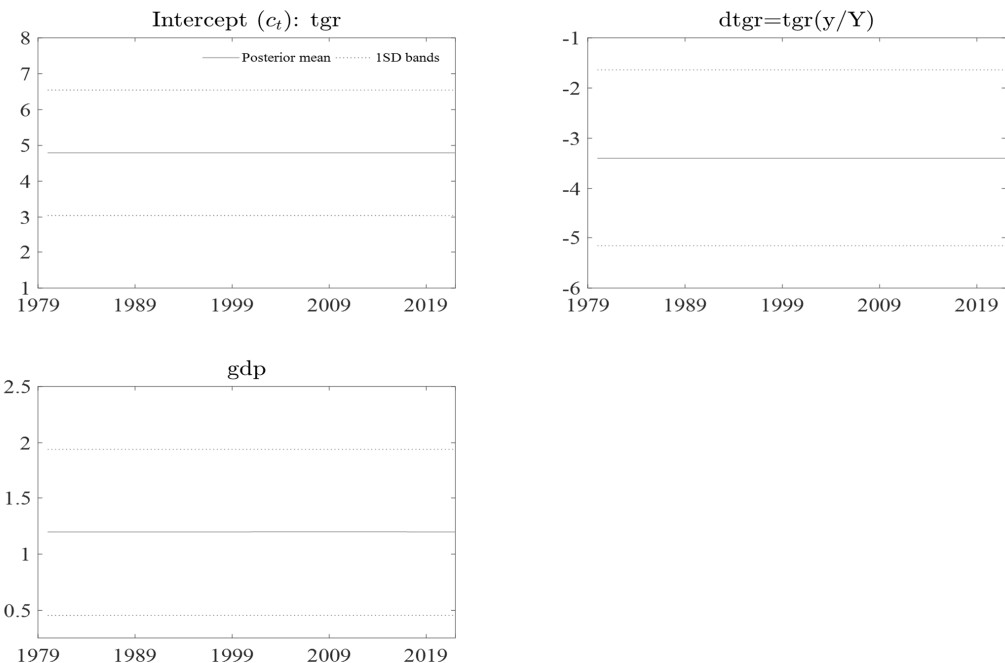

**Figure A9.** Graph is the evolution sequence of structural information for government revenue. Where $tgr$ is the total government revenue $tgr = g(y/Y)$ is the g total government revenue times the proportion of potation gross domestic product and $gdp$ is the gross domestic product per person. Composed by the authors.

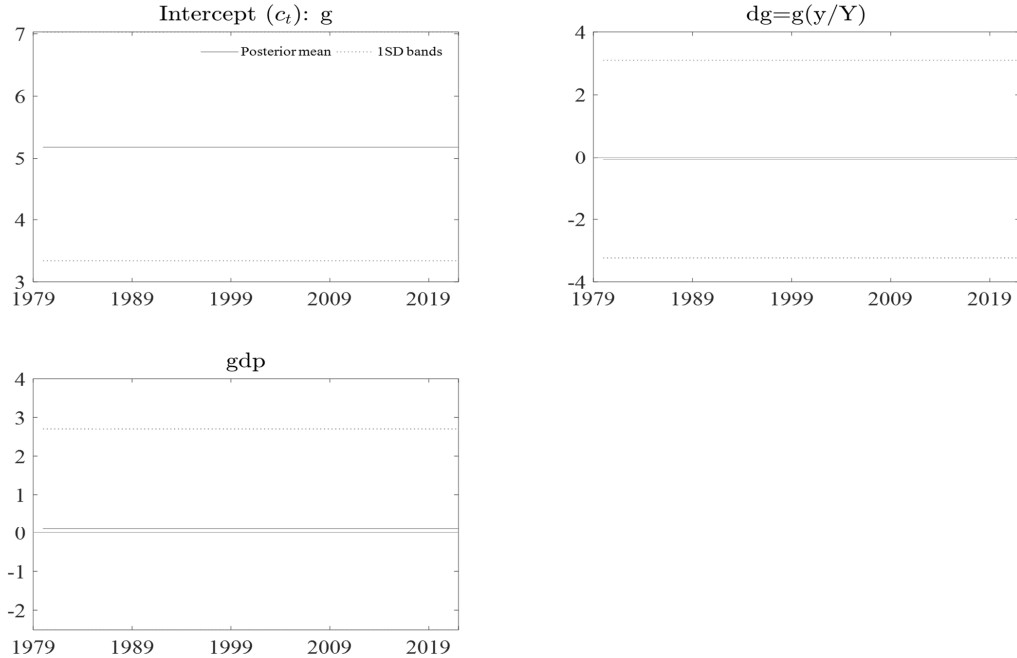

**Figure A10.** Graph is the evolution sequence of structural information for government expenditure. Where $g$ is the government expenditure $dg = g(y/Y)$ is the government expenditure times the proportion of potation gross domestic product and $gdp$ is the gross domestic product per person. Composed by the authors.

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
