# Peer review of "Time-Varying Elasticity of Cyclically Adjusted Primary Balance and Effect of Fiscal Consolidation on Domestic Government Debt in South Africa"

_economies, doi:10.3390/economies11050141_

Round 1
Reviewer 1 Report
The studied theme is very interesting because focuses on the cyclical adjusted primary balance (CAPB).
Major comments
1. The abstract is not clear regarding the methodology.
2. There are errors in every reference used in the paper, which makes it difficult to understand who the authors of the previous researches are, even the literature review is adequate to the subject. The novelty of research is clearly presented. Research hypotheses were clearly stated, and the results were correlated with them.
3.The methodology is accurately described. The results are detailed described.
Minor comments
1.I suggest ensuring English one more time.
2.The manuscript should respect more accurate the Journal template and to eliminate errors related to references.
Author Response
Good Day
Kindly find the attached document reflecting the comments, all changes are on track changes.
Thanks
Msizi

Reviewer 2 Report
Thanks for the opportunity to review this well-written paper. I have two main comments to be addressed before acceptance and publication.
1. The recent literature on fiscal reaction function is not referenced. Please cite the following paper published in Sustainability by MDPI.
Nakatani, R. Fiscal Rules for Natural Disaster- and Climate Change-Prone Small States. Sustainability 2021, 13, 3135. https://doi.org/10.3390/su13063135
This paper found the non-linear relationship between primary balance and public debt level. Specifically, the author found that there is an inverse U-shaped relationship between debt and primary balance. An economic intuition behind this result is that countries can benefit from debt financing by increasing investment and enlarging the economic base with low debt, but the costs of repaying debt increase as the debt reaches a critical level. This topic is related to the contents of your paper, so I encourage the authors to discuss this viewpoint and include the abovementioned article in the reference.
2. There are so many link errors regarding the references in the paper. The authors need to delete or fix all these link errors in the next round of revision of the manuscript.
Author Response
Good Day
Kindly find the attached document with comments addressed and changes are on track changes.
Thanks
Msizi

Reviewer 3 Report
The paper is well-written and can be interesting for scientists choosing the methods of the cyclical adjusted primary balance calculation.
There are some errors in the presentation of references.
Author Response

(The authors gave the same response as above.)

Reviewer 4 Report
The authors have analyzed the impact of fiscal consolidation on domestic government debt in South Africa over the 1979-2022 period through TVP-VAR model and cointegration test.
There are serious problems of language and referencing. Therefore, the language and references of the paper are checked and corrected.
Abstract is like a results section. They should shorten the abstract and the abstract should include the significance of the research topic, sample, method, main findings.
The authors clearly write why fiscal consolidation matters for the countries in the introduction.
The authors what they searched in the paper (what is the difference between the CAPB with time-varying elasticity and 75 time-invariant elasticity as a proxy for fiscal consolidation? The second is the impact of 76 fiscal consolidation on government debt in South Africa). But they clearly point out why they research taking notice of the related literature.
They should extend the literature about the relationship between fiscal consolidation and domestic government debt.
The authors should explain their results considering suggestions and standards by IMF and findings of similar studies. They also consider the crises between 1979-2022 while explaining their findings.
The authors should add the limitations of the study and give future research directions.
Author Response

(The authors gave the same response as above.)

Reviewer 5 Report
Time-Varying Vector Autoregressive model is utilized to estimate the parameters for South Africa’s primary balance.
Theoretical background is sufficiently provided in the text. Different types of approaches for measuring the balances are given.
It writes “s (Error! Reference source not found. to Error! Reference source not 282 found.).” on line 282.
Elasticities are calculated for both time invariant and time varying and they are compared for South Africa.
Author Response

(The authors gave the same response as above.)

Round 2
Reviewer 2 Report
Thanks for incorporating my comments. The manuscript became better. I recommend acceptance for publication, although English needs to be improved. Please use MDPI's or any other professional English editing service before publishing the final version of the manuscript.
Reviewer 4 Report
The authors have considerably improved the paper. Therefore, I suggest "accept in present form."